# Aggregation pheromones have a non-linear effect on oviposition behavior in *Drosophila melanogaster*

Thomas A. Verschut[1,2], Renny Ng[3], Nicolas P. Doubovetzky[1], Guillaume Le Calvez[4], Jan L. Sneep[4], Adriaan J. Minnaard [4], Chih-Ying Su [3], Mikael A. Carlsson [2], Bregje Wertheim [1] & Jean-Christophe Billeter [1] ✉

Female fruit flies (*Drosophila melanogaster*) oviposit at communal sites where the larvae may cooperate or compete for resources depending on group size. This offers a model system to determine how females assess quantitative social information. We show that the concentration of pheromones found on a substrate increases linearly with the number of adult flies that have visited that site. Females prefer oviposition sites with pheromone concentrations corresponding to an intermediate number of previous visitors, whereas sites with low or high concentrations are unattractive. This dose-dependent decision is based on a blend of 11-*cis*-Vaccenyl Acetate (cVA) indicating the number of previous visitors and heptanal (a novel pheromone deriving from the oxidation of 7-Tricosene), which acts as a dose-independent co-factor. This response is mediated by detection of cVA by odorant receptor neurons Or67d and Or65a, and at least five different odorant receptor neurons for heptanal. Our results identify a mechanism allowing individuals to transform a linear increase of pheromones into a non-linear behavioral response.

Proximity to conspecifics can confer benefits to individuals in overcoming constraints in survival and reproduction. These effects arise through cooperation in resource exploitation[1,2], predator avoidance through dilution effects[3,4], and increased chances of mating[5]. However, aggregation can also impose costs through competition for resources and increased risks of predation or pathogen transmission[3,6,7]. Consequently, the net effect for an individual of being in a group depends on the balance between the availability of resources, the benefits offered by cooperators, and the risks posed by competitors and natural enemies[8,9]. As these effects will often depend on density, it should be expected that there would be selection for mechanisms enabling individuals to evaluate local group size. Surprisingly, the mechanisms through which individuals assess group size are still poorly understood.

An illustration of the importance of group size comes from female insects searching for oviposition sites to lay their eggs. Selecting a suboptimal oviposition site may constrain offspring development due to the lack of nutrition[10,11], lead to increased intra- and interspecific competition for resources[12,13], or incur the risk of detection by natural enemies[14,15]. Ovipositing females have to make quantitative assessments of potential oviposition sites, balancing both the nutritional value and the costs and benefits of sharing the site with other females. Many insect species, including the fruit fly *Drosophila melanogaster* Meigen (Diptera: Drosophilidae), group their eggs at communal sites, which increases the probability that the larvae survive and complete their development[16–19]. Communal oviposition enhances oviposition site quality, both by the inoculation by the adults of yeasts, acting as a larval food source, and because groups of larvae are better at reducing the hyphal growth of molds that compete for food with the larvae[20–22].

[1]Groningen Institute for Evolutionary Life Sciences, University of Groningen, Nijenborgh 7, 9747 AG Groningen, The Netherlands. [2]Department of Zoology, Stockholm University, 106 91 Stockholm, Sweden. [3]Neurobiology Section, Division of Biological Sciences, University of California, San Diego, La Jolla, CA 92093, USA. [4]Stratingh Institute for Chemistry, University of Groningen, Nijenborgh 7, 9747 AG Groningen, The Netherlands. ✉e-mail: j.c.billeter@rug.nl

However, strong resource competition[23,24] and increased attraction of natural enemies may occur when groups of larvae become too large[14,25]. Modeling of population persistence and female decision making, based on behavioral and fecundity data of *D. melanogaster* females, predicts that substrate quality and adult group density are the main drivers for aggregation and communal oviposition[26,27]. Based upon these findings, it is expected that positive density-dependent effects occur when groups of larvae are neither too small nor too large[8,9]. Hence, females would benefit from assessing the number of females who have already contributed to a communal oviposition site before deciding on adding their own eggs to that site. A mechanism for this would be the ability to sense the dose of chemical cues left by contributors at the communal oviposition site to estimate whether a positive density-dependent effect on larval fitness may occur.

Pheromones are expected to play a key role in this mechanism as they instigate aggregation for feeding and ovipositing in many different species[1,28–30], advertise the presence of conspecifics to aid reproduction[31–34], and are used for communally fending off threats[35–37]. As pheromones have similarly been found to play a prominent role in the aggregation of *D. melanogaster*[31,38], we hypothesize that they may also be able to indicate the number of flies that have visited a communal oviposition site. The formation of a communal oviposition site begins when flies start visiting a fruit, where they are generally expected to spend at least an hour as the number of hourly arrivals on a particular fruit are normally greater than the number of departures[39]. Over the course of days, multiple females may visit the same site, and oviposition occurs both by females that concurrently and sequentially visits the site[40]. While visiting a fruit, males and mated females leave behind deposits of 11-*cis*-vaccenyl acetate (cVA)[20,41] and several cuticular hydrocarbon pheromones, including the male dominant monoalkenes, (Z)−7-tricosene (7-T) and (Z)−9-tricosene (9-T), and the female specific diene (7Z,11Z)-heptacosadiene (7,11-HD)[18,42,43]. These pheromones affect the oviposition behavior of other females after the depositor has left the substrate[18,40,41,43–45]. Whether the cuticular hydrocarbon pheromones act at a distance is unclear as these hydrocarbons are typically assumed to function as contact cues sensed by the gustatory system[46,47]. On the other hand, cVA volatiles are sensed by the olfactory system and attract *Drosophila* when sensed in combination with food odors[17,41,48–50]. Understanding the complexity of sensory cues at play asks for a dissection of the mechanisms through which individual flies assess the dose of pheromones and adjust their behavioral responses accordingly.

In this study, we investigate whether females use pheromones as quantitative cues to determine the suitability of communal oviposition sites. We hypothesize that the females' oviposition decisions are modulated by the dose of volatile pheromones, as these provide an indication for the number of individuals that have contributed to that site. Since larval survival at communal oviposition sites follows a hump-shaped relationship with larval density, in which sites with too few or too many larvae will impose developmental constraints[9,20,22,51], we hypothesize that the attraction to pheromonal deposits will increase up to an optimum and then decrease. Given that flies assess both social and nutritional cues when determining the quality of an oviposition sites for larval survival[18], we expect that the decision to oviposit at a communal oviposition site is based on cues from previous visitors to the communal site and cues about the nutritional quality. We also expect that this decision should be made through volatiles sensed prior to laying eggs on the site. Therefore, we focus on olfactory cues and not on gustatory cues, which would necessitate micro-assessment of the substrate after arrival on the substrate[46,47,52]. Finally, considering that the occurrence of positive density-dependent effects are expected to depend on resource conditions in relation to group size[8,9,18,26], we hypothesize that females rely less on pheromonal cues when evaluating oviposition substrates of high nutritional quality than on substrates of low nutritional quality, since larvae depend more on cooperative behavior to survive on poor nutrition substrates[20].

Here, we show that females respond to pheromonal extracts of males and mated females following a non-linear inverted U-shaped dose-response curve. This dose-response curve is generated by co-sensing cVA and heptanal, a novel pheromone produced by the oxidization of 7-T. The determination of these quantitative cues depends on the involvement of Or67d and Or65a, at different concentrations of cVA, and the activation of multiple olfactory receptors by heptanal. Consequently, our data reveal a pheromonal mechanisms through which an individual female can evaluate an oviposition site before making reproductive decisions.

## Results

### Females show a non-linear behavioral response to pheromone dose when selecting oviposition sites

We developed a two-choice olfactory oviposition assay to test the hypothesis that females use volatile pheromone concentrations to select oviposition sites that are communally used by conspecifics. The assay consists of two soft agar oviposition zones separated by a hard middle zone unsuitable for oviposition. The agar of the oviposition zones enclosed a small mesh-covered cup into which pheromones could be loaded. This ensured that pheromones could only be sensed as volatile indicators, and could not be detected through physical contact (Fig. 1A). Prior to testing the dose-response to increasing concentrations of pheromones on females' oviposition site selection, we determined whether the total concentration of pheromones deposited by individuals increases with group size. We kept one, two, six, and twelve $w^{1118}$ males in small glass tubes for 90 min, to mimic a natural visit to a substrate[39], and quantified the pheromonal deposits left behind on the glass. We also randomly selected a single male from each of those groups and quantified the pheromones present on the cuticle of that single fly (Table S1). We found an interaction between deposited pheromones and pheromones on the cuticle of a single male with increasing group size (GLM: $\chi^2_{3,60} = 32.11$, $P < 0.001$; Fig. 1B). More specifically, the concentration of pheromones on the cuticle is unaffected by group size (GLM: $\chi^2_{3,28} = 1.34$, $P = 0.719$), whereas the concentration of deposited pheromones increases linearly with group size (GLM: $\chi^2_{3,28} = 60.12$, $P < 0.001$; Fig. 1B). These data show that quantity of deposited pheromones increases linearly with the number of flies present on a substrate. Therefore, the concentration of pheromones at a site can serve as a reliable proxy for the number of flies that visited that site.

The effect of pheromonal dose, representing the number of flies that deposited pheromones at the oviposition site, was then tested using increasing doses of full-body fly pheromone extracts of either male or mated female single-sex groups (see Methods). We focused on males and mated females since it was previously shown that virgin females do not deposit oviposition inducing pheromones[18]. As full body extracts over-represent the concentration of pheromones a fly deposits on a substrate (Fig. 1B), we also tested several smaller fractions of the pheromone extracts. In addition, to test the hypothesis that the response to pheromone concentration depends on the nutrient quality of the location, we ran experiments in which the oviposition zones either contained only sucrose, or sucrose combined with yeast (see Methods). Oviposition sites only containing sucrose offer a low-quality resource that is expected to be too poor for sustaining larval development. This can be mediated when multiple females communally inoculate the oviposition site with yeasts during oviposition. The addition of yeast to the oviposition zones should therefore support the development of groups of larvae and reduces the need for communal oviposition[26,53].

Using this assay, we found that mated females respond differently to increasing doses of pheromones depending on an interactive effect

between food conditions and the sex from which the pheromonal extracts originated (GLM: $F_{1,795} = 6.41$, $P = 0.011$; Fig. 1C, D; Tables S2–S3). More specifically, when the oviposition substrates only contained sucrose, the response to pheromone concentration followed a non-linear pattern, with attraction to a quarter male extract up to extracts of three males (GAM: $F_{1,391} = 6.52$, $P < 0.001$), and a quarter mated female extract up to extracts of four mated females (GAM: $F_{1,391} = 6.05$, $P < 0.001$; Fig. 1C; Tables S2–S3). However, when the substrate contained sucrose and yeast, the response to increasing doses of pheromones decreased linearly and were affected by sex and group size, with attraction up to extracts of two mated females (GLM: $F_{1,198} = 7.99$, $P = 0.005$), and only up to half a male extract (GLM: $F_{1,198} = 23.87$, $P < 0.001$). The largest pheromone doses representing six and twelve males even repelled the females from ovipositing on the site marked by those quantities of pheromones (Fig. 1D; Tables S2–S3). The outcome of this experiment suggests that mated females modulate oviposition site selection according to the concentration of pheromones present at an oviposition site, and select communal oviposition sites depending on cues that reveal the number of flies that

have visited that site. Furthermore, the results show that the preference to oviposit at a site containing different doses of pheromones depends on the food conditions at the site.

## Male pheromones are necessary to attract females to communal oviposition sites

To determine which specific pheromonal compounds are involved in the pheromone dose-dependent selection of oviposition sites, we continued our experiments with assays devoid of yeast to remove nutritional quality as a variable. The first step was to determine from which sex the attractive pheromones originated, since the pheromonal profile of males and mated females share similarities following the exchange of pheromones during copulation (Fig. 2A)[54,55], and no differentiation is made between oviposition sites visited by male or mated female pheromonal extracts when given the choice between the two (Fig. S1A). To that purpose, we ablated the oenocytes (Oe⁻), cells that produce cuticular hydrocarbons, to acquire flies that are devoid of these pheromones[56]. While we found no attraction to extracts of Oe⁻ males, Oe⁻ virgin females or Oe⁻ females mated to Oe⁻ males, we found

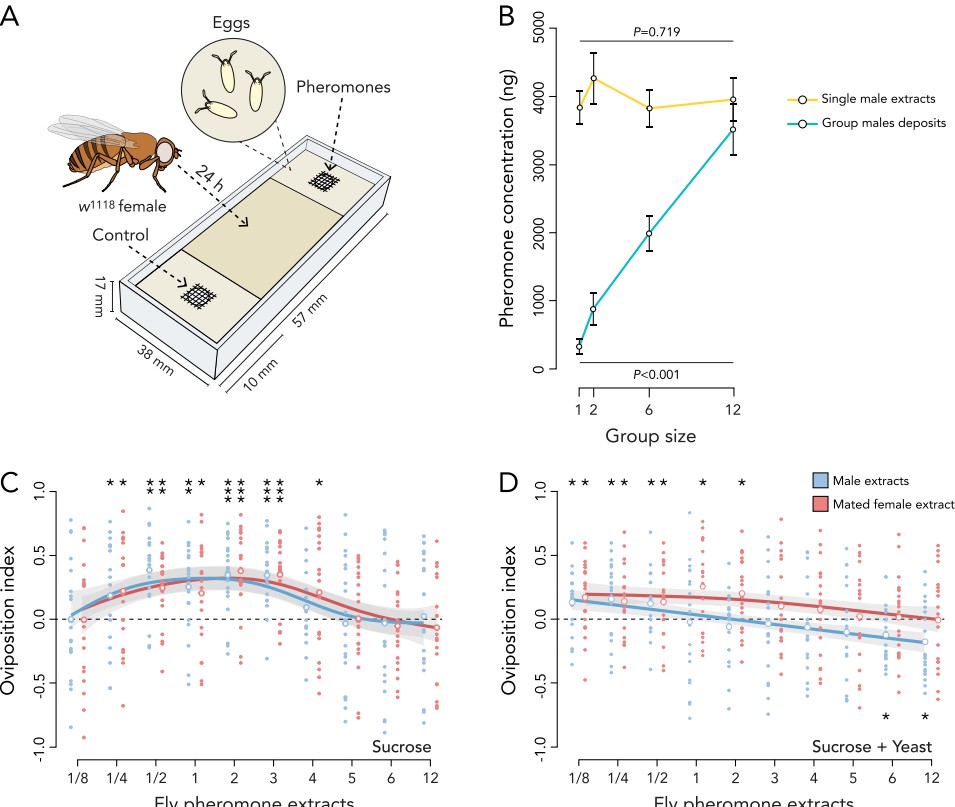

**Fig. 1 | Pheromones affect communal oviposition site selection in a non-linear dose dependent manner. A** Overview of the two-choice oviposition assay. The female can lay eggs in oviposition zones of 0.75% agar at either end of the assay (light zones). Both oviposition zones enclose a mesh covered cup containing the pheromones or solvent control treatments. These two oviposition zones are separated by a zone of 3% agar unsuitable for oviposition (dark zone). After 24 h, eggs are counted and an oviposition index is calculated: (Eggs side 1 − Eggs side 2) / (Eggs side 1 + Eggs side 2). **B** Cuticular pheromone extracts of single $w^{1118}$ males kept in groups of 1, 2, 6 or 12 males (yellow) and the total concentration of pheromones deposited by these groups of $w^{1118}$ males (turquoise). The concentrations were compared using a GLM and the error bars indicate the standard error of the mean. The line and $P$-value above the plotted lines represents the analysis of the single male extracts and the line and $P$-value below the plotted lines represents the analysis of the deposits made by grouped males. Table S1 reports the total pheromone concentrations and that of individual compounds ($n = 9$ for 1 fly, $n = 8$ for 2 flies,

$n = 6$ for 6 flies and $n = 8$ for 12 flies). **C** Oviposition preference to pheromone extracts of increasing numbers of males (blue) or mated females (red) on oviposition zones containing 100 mM sucrose (see methods). **D** Oviposition preference on oviposition zones containing 100 mM sucrose and 8.75 g/L of yeast. The individual replicates are visualized by the small data points and the mean oviposition indices are given by larger unfilled data points. These data points are analyzed using two-tailed Wilcoxon signed rank tests. The non-linear analysis (fitted with GAM) in **C** and linear analysis (fitted with GLM) in **D** are visualized with their 95% confidence intervals (gray shaded area). The asterisks above or below a treatment respectively indicate attraction or aversion differing significantly from zero (0.0 dashed line− indicating no preference) as determined by two-tailed Wilcoxon signed rank tests ($n = 20$ for all treatments). *$P < 0.05$; **$P < 0.01$; ***$P < 0.001$. No asterisks indication means no significant difference from zero. See Fig. S1D for solvent control treatments and Tables S1–S3 for the full outcome of the statistical analyses. Source data are provided with this paper.

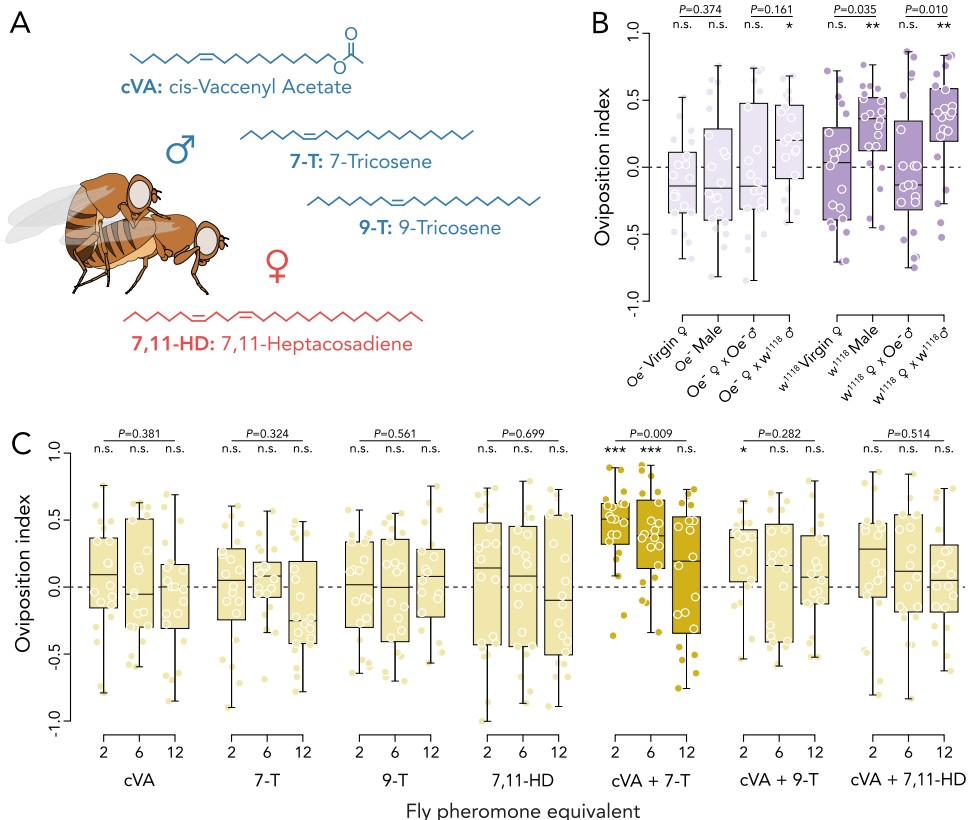

**Fig. 2 | Male-derived pheromones attract mated females to oviposition sites.** **A** Overview of the main sex-specific pheromones transferred by males (blue) and females (red) during mating. **B** Oviposition preference to pheromone extracts of oenocyte ablated (Oe⁻), $w^{1118}$, or mated flies of the indicated genotypes (see Fig. S1A−B for controls). **C** Oviposition preference to two, six and twelve times the single fly pheromonal equivalent of *cis*−11-vaccenyl acetate (cVA: 560−1680−3360 ng), 7-tricosene (7-T: 280−840−1680 ng), 9-tricosene (9-T: 28−84−168 ng; see Fig. S1C) and 7,11-heptacosadiene (7,11-HD: 620−1860−3360 ng) and combinations thereof. Asterisks above a treatment indicates attraction differing significantly from zero

(0.0 dashed line−indicating no preference) as determined by two-tailed Wilcoxon signed rank tests ($n = 20$ for all treatments). Differences in preference across treatments was tested with a GLM with quasibinomial error distribution and is indicated above the bar. *$P < 0.05$; **$P < 0.01$; ***$P < 0.001$. No asterisks means no significant difference from zero for the Wilcoxon signed rank tests. The center line of the box plots denotes the median value (50th percentile), the box contains the 25th to 75th percentiles of dataset. The black whiskers mark 1.5 times the inter-quartile range. See Fig. S1D for solvent control treatments and Tables S4−S7 for the full outcome of the statistical analyses. Source data are provided with this paper.

attraction to extracts of Oe⁻ females that had mated with wild type males (Fig. 2B). As the non-attractive Oe⁻ males are still able to produce cVA, this result suggests that a combination of cVA and cuticular hydrocarbons is needed to attract females to communal oviposition sites. To validate this hypothesis, we tested attraction to extracts from individual flies of both sexes and all combinations of possible mated genotypes, and only found attraction to males producing both cVA and cuticular hydrocarbons, and females that had mated with wild-type males (Fig. 2B; Fig. S1B; Tables S4−S5). In line with previous findings by Duménil et al. (2016)[18], this shows that both cVA and male specific cuticular hydrocarbons are necessary pheromonal components to attract mated females to communal oviposition sites.

We next aimed to identify which specific combination of compounds underlies the inverted U-shaped pheromone dose-response for oviposition site selection. We selected four compounds that vary in concentration between virgin and mated females[54] as the most likely candidates involved in the behavioral response (Fig. 2A): cVA, the male dominant 7-T and 9-T, and the female-specific 7,11-HD. Of these four compounds, cVA is the male-derived aggregation pheromone[41], and 7-T is the major male pheromone transferred to females upon mating[54]. Furthermore, 9-T deposits have been reported to stimulate females to lay eggs[43], and volatile breakdown products of 7,11-HD have been found to act as a long-range attractant[57]. We hypothesized that if any of these compounds are dominantly involved in oviposition site selection, they should reproduce the

decrease in attraction following the exposure to increasing doses of fly pheromone extracts (Fig. 1). We used pheromone equivalents, corresponding to two, six, and twelve times the amounts of cVA, 7-T, 9-T, and 7,11-HD found on a single fly to test if they could prompt a dose-dependent decrease in attraction with increasing concentrations as observed with whole fly extracts (Fig. 1C). Neither cVA, nor any of the cuticular hydrocarbons attracted mated females by themselves (Fig. 2C). As it has been previously documented that cVA functions synergistically with 7-T[54], we tested all three cuticular hydrocarbons in combination with cVA to determine whether a combination of pheromones can act as a group size indicator. We found a dose-dependent decrease in attraction to the combination of cVA and 7-T (GLM: $F_{1,57} = 6.83$, $P = 0.009$; Fig. 2C). For the combination of cVA and 9-T, we only found attraction to the lowest pheromone equivalents (Fig. 2C), but no dose-dependent decrease (GLM: $F_{1,57} = 1.56$, $P = 0.282$; Fig. S1C; Tables S6−S7). Finally, combining cVA with the female-specific 7,11-HD resulted in no dose-dependent decrease in attraction (GLM: $F_{1,57} = 0.43$, $P = 0.514$; Fig. 2C; Tables S6−S7). These results suggest that cVA and 7-T are the two predominant compounds that instigate oviposition in a concentration-dependent manner. Moreover, these results suggest that the combination of cVA and 7-T is sensed by olfaction and not taste, as a mesh prevented the females from making contact with the pheromones in the assay. Therefore, these two compounds can serve as an indicator of group-size dependent oviposition site quality at a

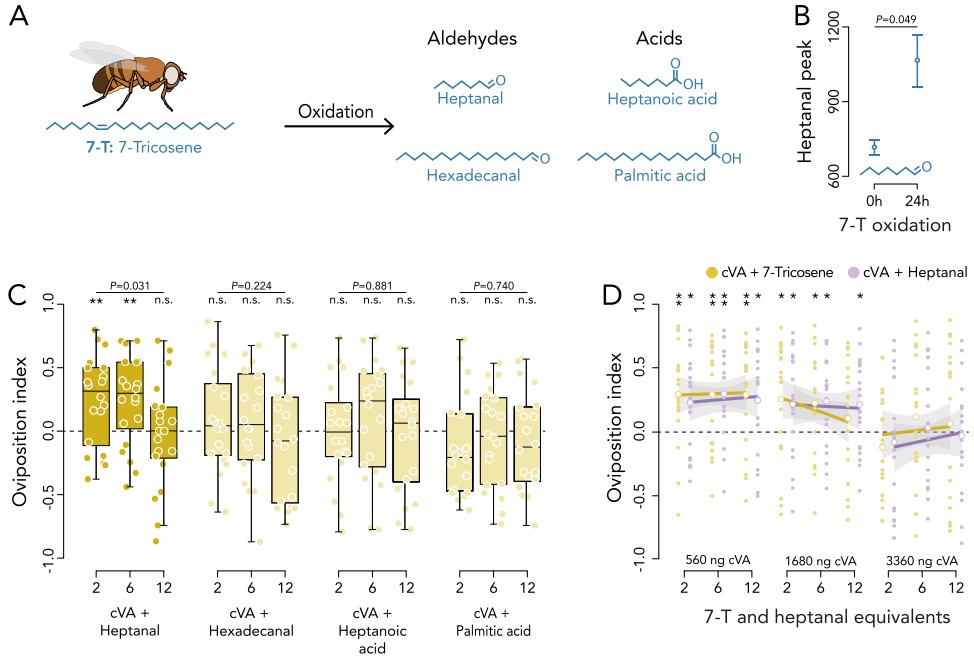

**Fig. 3 | The oxidation of 7-Tricosene and its effect on the detection of oviposition sites. A** 7-Tricosene is predicted to autoxidize into heptanal and hexadecanal (both saturated aldehydes), heptanoic acid (an aliphatic carboxylic acid) and palmitic acid (a saturated long-chain fatty acid). **B** Mean area under the peak for heptanal directly (i.e. 0 h) and 24 h after exposing 7-T to ambient oxygen. Statistical outcome as analyzed with a Kruskal–Wallis test. Error bars indicate standard error of the mean ($n = 3$ per time point). **C** Oviposition preference to cVA combined with heptanal (100–300–600 ng), hexadecanal (210–630–1260 ng), heptanoic acid (115–345–690 ng) and palmitic acid (220–660–1320 ng; Fig. S1E–F for additional treatments). **D** Oviposition preference to two, six and twelve times the single fly pheromone equivalent of cVA (560–1680–3360 ng) combined with each concentration of 7-T (ochre: 280–840–1680 ng) or heptanal (purple: 100–300–600 ng). The linear analysis (fitted with GLM) is visualized with its 95% confidence intervals (gray shaded area). The individual replicates are visualized by the small data points and the mean oviposition indices are given by larger unfilled data points. These data points are analyzed using two-tailed Wilcoxon signed rank tests. Asterisks above a treatment indicate attraction differing significantly from zero (0.0 line—indicating no preference) as determined by the two-tailed Wilcoxon signed rank tests ($n = 20$ for all treatments). Differences in preference across treatments were tested with a GLM with quasibinomial error distribution and is indicated above the bar when applicable. *$P < 0.05$; **$P < 0.01$; ***$P < 0.001$. No asterisks indication or n.s. means no significant difference from zero for the Wilcoxon signed rank tests. The center line of the box plots denotes the median value (50th percentile), the box contains the 25th to 75th percentiles of dataset. The black whiskers mark 1.5 times the interquartile range. Data points beyond these bounds are considered outliers. See Fig. S1D for solvent control treatments and Tables S8–S11 for the full outcome of the statistical analyses. Source data are provided with this paper.

distance, without the assessment of deposited pheromones through additional sensory systems.

## 11-*cis*-vaccenyl acetate (cVA) is a quantitative cue that requires the co-occurrence of heptanal

The involvement of 7-T as a volatile cue during oviposition site selection challenges the general assumption that cuticular hydrocarbons have limited volatility and are detected through gustatory contact sensilla[58–60]. Several studies have, however, suggested that flies may detect cuticular hydrocarbons through their olfactory system[42,43,55,61]. It is also reported that the oxidation of 7,11-HD at its omega-7 and −11 double bonds by ambient oxygen produces several short aldehydes, of which (*Z*)−4-undecenal attracts flies over long distances[57]. We hypothesized that 7-T, which contains one omega-7 double bond, may also oxidize into smaller, more volatile, molecules. The predicted autoxidation products of 7-T include the saturated aldehydes heptanal and hexadecanal, which both have been documented in the headspace of flies[57], heptanoic acid (an aliphatic carboxylic acid) and palmitic acid (a saturated long-chain fatty acid) (Fig. 3A). While none of the four oxidation products induce communal oviposition by themselves (Fig. S1E–F), the combination of heptanal and cVA resulted in a dose-dependent decrease with increasing pheromone concentrations (GLM: $F_{1,57} = 4.69$, $P = 0.031$; Fig. 3C; Fig. S1F; Tables S8 and S9), similar to the response to cVA and 7-T (Fig. 2C). None of the other candidate compounds had an effect in combination with cVA (Fig. 3C; Fig. S1E;

Tables S8 and S9). This suggests that heptanal is the volatile oxidation product of 7-T that causes attraction. To validate that heptanal is an autoxidation product of 7-T, we performed headspace analysis and found that 7-T immediately starts oxidizing into heptanal upon contact with ambient oxygen (Fig. 3B). Moreover, 7-T samples exposed to ambient oxygen for 24 h contained 48% more heptanal in their headspace than samples directly analyzed upon exposure to ambient oxygen (Kruskal–Wallis: $H_1 = 3.86$, $P = 0.049$; Fig. 3B). This validates that heptanal is released following the oxidation of 7-T by ambient oxygen and corroborates the assumption that cuticular hydrocarbons can affect fly behavior through their volatile oxidation products, without the need for direct physical contact.

We next wanted to determine which of the three compounds, cVA, 7-T or heptanal, predominantly functions as the quantitative cue. To that purpose, we tested oviposition attraction to all combinations of these compounds at two, six and twelve times their fly equivalents. We observed a dose-dependent decrease in oviposition attraction with the increasing concentration of cVA (GLM: $F_{1,356} = 31.65$, $P < 0.001$), irrespectively of it being combined with 7-T or heptanal, and irrespective of 7-T or heptanal dose (GLM: $F_{1,356} = 0.37$, $P = 0.543$; Fig. 3D; Tables S10 and S11). These results suggest that females use cVA concentration as a dose-dependent indicator of group size, and that 7-T and heptanal interchangeably function as dose-independent co-factors attracting females to communal oviposition sites. We postulate that 7-T and heptanal act as a recognition cue of sites where males or

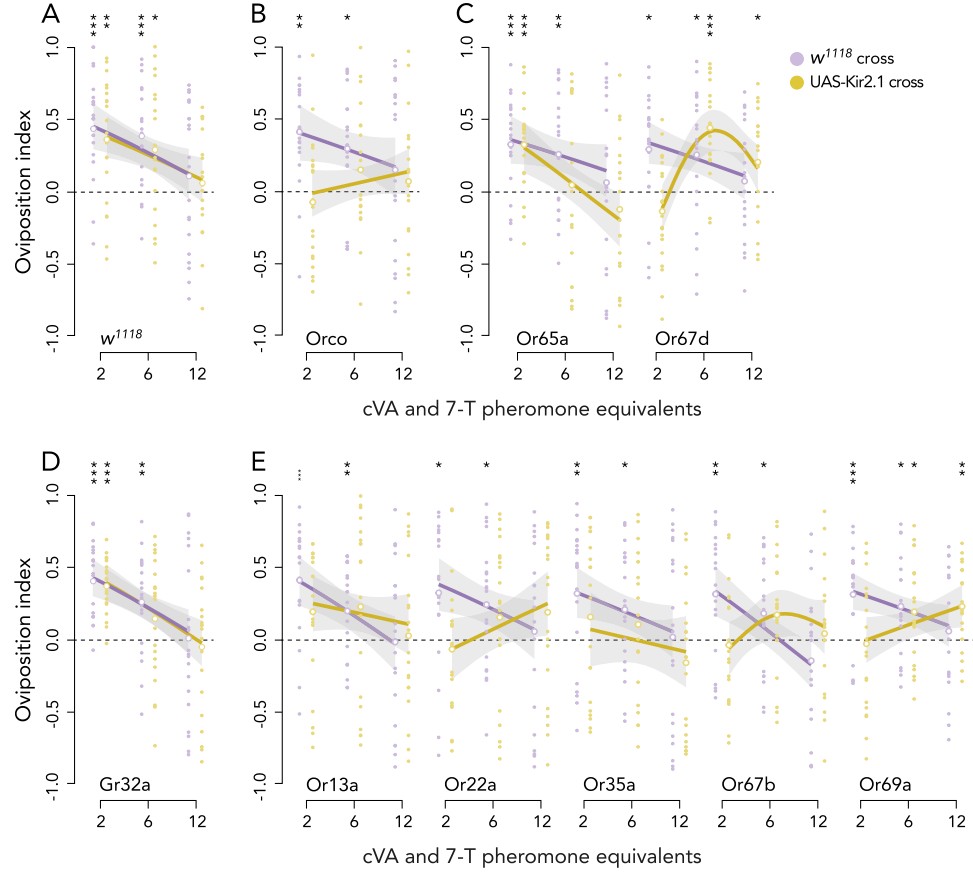

**Fig. 4 | Candidate receptors involved in the dose-dependent detection of cVA and 7-T or heptanal. A** Oviposition preference to two, six and twelve times the single fly pheromone equivalent of cVA (560– 1680–3360 ng) combined with 7-T (280–840–1680 ng) for $w^{1118}$ control females crossed to $w^{1118}$ (purple) or crossed to *UAS-Kir2.1* (ochre). Oviposition preference of control females and females with silenced receptors for **B** Orco; **C** olfactory receptors Or65a and Or67d for cVA; **D** gustatory receptor Gr32a for 7-T; **E** olfactory receptors Or13a, Or22a, Or35a, Or67b, and Or69a for heptanal. The non-linear analysis (fitted with GAM) and linear analysis (fitted with GLM) are visualized with their 95% confidence intervals (gray shaded area). The individual replicates are visualized by the small data points and the mean oviposition indices are given by larger unfilled data points. These data points are analyzed using two-tailed Wilcoxon signed rank tests. Asterisks above each treatment indicate whether preference to the pheromone treatments differ significantly from zero (0.0 line–indicating no preference) as determined by the two-tailed Wilcoxon signed rank tests ($n = 20$ for all treatments). Difference in preference across treatments was tested with a GLM or GAM with quasibinomial error distribution and is indicated above the bar. *$P < 0.05$; **$P < 0.01$; ***$P < 0.001$. No asterisks indication or n.s. means no significant difference from zero for the Wilcoxon signed-rank tests. See Fig. S1D for solvent control treatments and Tables S12–S13 for the full outcome of the statistical analyses. Source data are provided with this paper.

mated females are present. We base this on the finding that virgin females, who contain minimal amounts of 7-T[54], do not attract other females. This recognition is functional as virgin females are not a reliable signal of oviposition site quality since they cannot contribute fertilized eggs[62,63].

## Concentrations of *cis*−11-vaccenyl acetate (cVA) are determined by odorant receptors Or65a and Or67d

As previous studies indicated that the classical olfactory receptors (Or)[64,65], rather than the ionotropic receptors (Ir)[18,66], are necessary for detecting pheromones deposited at communal oviposition sites[18], we silenced all olfactory receptor neurons (ORN) by targeting the expression of the inwardly rectifying potassium channel Kir2.1 in *Orco*-expressing neurons. Compared to wild-type flies (Fig. 4A), the ORN-silenced flies showed no oviposition preference in response to any concentrations of cVA and 7-T (Fig. 4B; Tables S12–S13), showing that *Orco*⁺ ORNs are necessary for communal oviposition site selection. Next, we determined the involvement of specific ORN types in the group size-dependent response to cVA and 7-T. It has been suggested that cVA is detected by Or67d and Or65a, of which Or67d is involved in acute responses, and Or65a presumably modulates the response during longer exposure to cVA by inhibiting interneurons receiving

input from Or67d[48,67,68]. Based on the different response properties of Or67d and Or65a, we postulate that they may respond to different concentrations of cVA and generate distinct parts of the inverted U-shaped dose-response curve. When we silenced the two types of cVA receptor neurons, we found that silencing Or65a ORNs lowered the attraction to cVA and 7-T at the two lowest doses but not at the highest dose (Fig. 4C). Silencing Or67d ORNs resulted in a non-linear response with a lack of attraction towards the lowest dose and a slight increase in attraction compared to the controls at six and twelve times the pheromone equivalent (Fig. 4C; Tables S12–S13). These results suggest that Or65a and Or67d may jointly regulate the sensing of different doses of cVA, and that they together determine the shape of the dose-response curve found for increasing doses of pheromone extracts.

## Multiple ORNs are necessary for dose-dependent response to 7-Tricosene (7-T)

Our results suggested that cVA indicates the number of individuals previously visiting a site in a dose-dependent manner, but needs the co-occurrence of 7-T (or its oxidation product heptanal) to attract mated females. As we already established that both Or67d and Or65a are involved in the dose-dependent response to cVA, we next wanted to determine which receptors mediate the dose-independent effect of

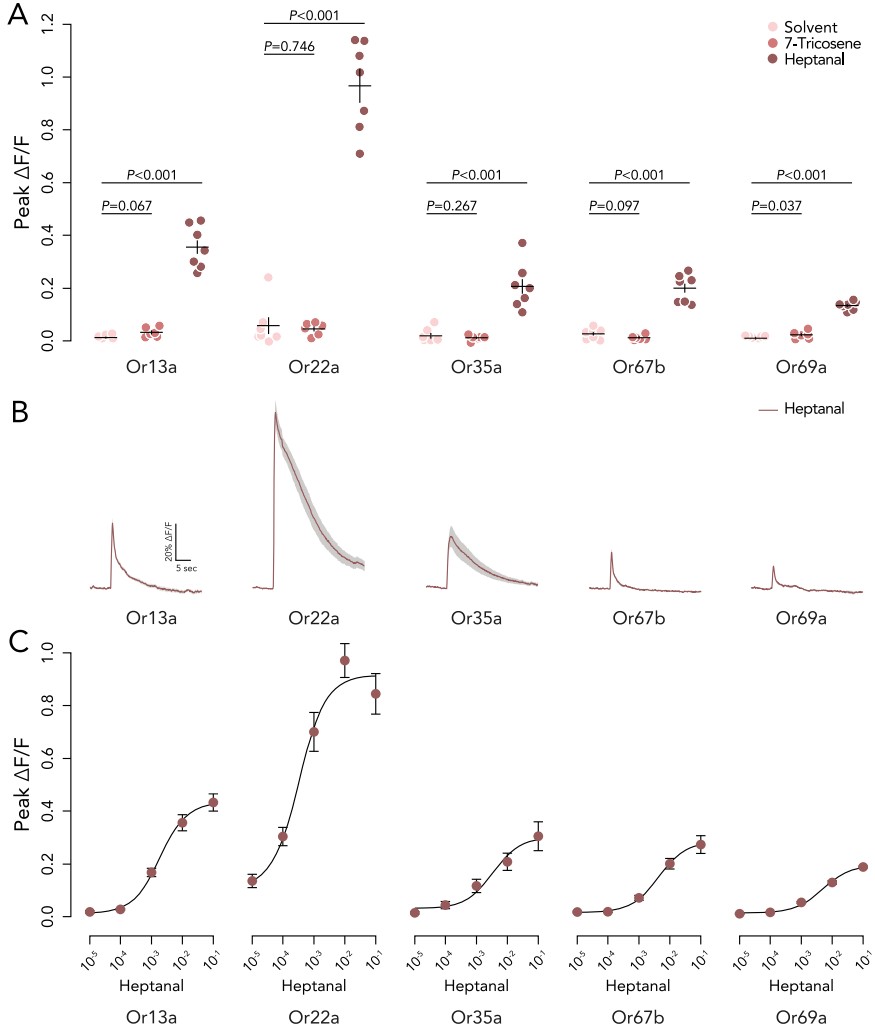

**Fig. 5 | Transcuticle calcium imaging of candidate olfactory receptor neurons for heptanal. A** Quantification (mean ± standard error of the mean) of transcuticle calcium responses from the indicated olfactory receptor neurons (ORN) to the paraffin oil solvent ($n = 7$), 7-Tricosene ($n = 6$), or heptanal ($n = 7$). Statistical differences between the treatments were calculated with two-tailed $t$-tests comparing the responses to solvent and 7-Tricosene and solvent and heptanal ($10^{-2}$ dilution).

**B** Calcium fluorescence traces of the candidate ORNs in response to a $10^{-2}$ dilution of heptanal visualized with standard error of the mean as gray shaded areas. **C** Dose-response curves of the candidate ORNs to aliquots of heptanal ($n = 7$ per aliquot). The error bars indicate the standard error of the mean. See Fig. S2 for negative control on Or47b. Source data are provided with this paper.

7-T. The only documented receptor for 7-T is the gustatory receptor Gr32a[58], which is an unlikely candidate given that 7-T can only be detected as a volatile in our assay. To validate the assumption that females do not respond to 7-T by direct contact through the mesh covering the pheromone treatments, we silenced Gr32a⁺ neurons, but found that Gr32a-silenced females responded normally to increasing pheromone concentrations (Fig. 4D; Tables S12–S13). This validates the assumption that 7-T is not detected through taste receptors in the oviposition assay, and suggests that the groups size dependent response must rely on the volatile derivatives of 7-T.

To identify potential olfactory receptors for heptanal, we used the Database of Odorant Responses to identify olfactory receptors predominantly involved in the detection of heptanal by *D. melanogaster* (DoOR 2.0–[69]). As a selection criterion we only selected the olfactory receptors with a response value above 0.1. This resulted in the selection of Or13a (Response value: 0.257), Or22a (0.497), Or35a (reported as ac3B – 0.585), Or67b (0.158), and Or69a (0.199) as candidate receptors. We found that four of these, Or13a, Or22a, Or35a, and Or67b, are all necessary for attraction to cVA and 7-T-marked oviposition sites, as silencing neurons expressing these ORs reduced preference towards these pheromones (Fig. 4E; Tables S12–S13). This

suggests that these ORNs normally work in concert to detect heptanal and mediate attraction to a communal oviposition site. For Or69a, we note that its involvement is unclear as silencing Or69a only led to a lack of attraction at the lowest pheromone concentration and an increased attraction to the highest pheromone concentration compared to the control (Fig. 4E; Tables S12–S13).

Finally, to test the sensitivity of the five ORN types to 7-T and heptanal, we specifically expressed GCaMP7 in each of the ORNs and used transcuticular calcium imaging[70], to test neuronal activity in response to a single high dose of 7-T and several dilutions of heptanal. Of all ORNs tested, only Or69a showed a statistically significant response to 7-T (Fig. 5A). Considering that this response is barely above the background signal, we surmise that this response can only have a negligible effect on sensing 7-T during oviposition site selection. Instead, we expect that this small response to 7-T may be an effect of the rapid oxidation of 7-T into heptanal upon contact with oxygen (Fig. 3B), or an artifact of Or69a's dual affinity to pheromonal compounds and environmental semiochemicals[57]. However, all five ORNs responded to heptanal (Fig. 5A). The responses increased in a dose-dependent manner, with Or22a manifesting the strongest calcium response (Fig. 5B, C; Fig. S2 for negative control on Or47b). This

suggests that heptanal is sensed by a group of ORNs, whose co-activation indirectly signals the presence of 7-T, through heptanal, and allows attraction to pheromone-marked sites.

## Discussion

We found that *Drosophila melanogaster* females choose a potential oviposition site through the dose of volatile pheromones deposited by previous visitors, and that they select these sites in a quantitative manner, adjusted to the available food resources (Fig. 1). Specifically, the combination of the male derived 11-*cis*-Vaccenyl Acetate (cVA) and 7-Tricosene (7-T) functions as the dose-dependent cue. The observed dose-dependent pattern is consistent with the expectations for whether laying eggs at a site may result in positive- or negative density-dependent effects on larval development and survival. The non-linear behavioral response predominantly depends on the concentration of cVA, and involves odorant receptors Or67d and Or65a (Fig. 4C). Furthermore, it necessitates the co-occurrence of the dose-independent 7-T to instigate oviposition. This cuticular hydrocarbon of low volatility needs to oxidize into the more volatile heptanal (Fig. 3A, B), and it co-activates multiple olfactory receptors (Fig. 4E), before it can affect oviposition site selection. Because we tested the attraction of females towards oviposition substrates marked by pheromone extracts from different numbers of individuals instead of testing attraction to the individuals themselves, it is important to note that our results uncover a mechanism potentially allowing females to determine the number of prior individuals sharing a communal oviposition site through olfaction alone. Furthermore, our results identify a mechanism enabling individuals to transform a linear increase in pheromones into a non-linear behavioral response fitting the theoretically hypothesized costs and benefits of local density[8,9,20,51]. Because we aimed to specifically study the olfactory social component of attraction to oviposition sites, a limitation of our study is that we did not test attraction to different group sizes of adult flies, just the pheromones they can deposit. It is likely that additional social factors modulate oviposition site selection once flies have arrived at an oviposition site. For example, flies physically present at the oviposition site may modify oviposition behavior through visual input[71], physical interaction between individuals[72], or the release of stress pheromones[73,74].

### Volatile cues are used to select oviposition sites

Considering that beneficial density-dependent effects usually occur when groups are neither too small nor too large[8,9], females should be able to recognize cues that indicate optimal or suboptimal social conditions for their offspring[19,26]. In our experiments, volatiles deriving from the accumulation of cVA and heptanal, a volatile derivative of 7-T deposits from both males and mated females, could act as cues for whether an oviposition site offers optimal or suboptimal conditions. The biological relevance of the combination of cVA and 7-T may lie in the fact that these two pheromones are only associated with males and mated females[54,55]. More specifically, the presence of mated females is expected to be a relevant signal for oviposition site quality, as these mated females also require larval food sources containing yeast[53,75], and they provide the communal benefits by contributing both yeast and fertilized eggs at the shared site[18]. Males offer a relevant signal as they are necessary for fertilizing females[53,54,75]. We expect that cues of the presence of virgin females have little relevance given that they may introduce yeast, but they cannot contribute fertilized eggs[62,63], and that the fitness benefits of laying eggs communally predominantly comes from the number of larvae developing at the communal site[20,22,26]. This number of larvae should not be too small, in which the larvae fail to survive due to harmful fungal growth on the substrate, and not too large that strong resource competition leads to cannibalism. The combination of cVA and 7-T is thus a specific cue of the presence of flies that are good indicators of oviposition site quality and means

that mated females are not attracted to pheromones left behind by virgin females.

The combination of cVA and 7-T is also involved in mate-guarding, by making females resemble the pheromone profile of males, which inhibits courtship attempts by males through physical contact with taste receptor Gr32a[54]. Because of their use in sexual communication, females might preferentially lay eggs on sites containing intermediate amount of these pheromones, not because they are evaluating it as a preferred place to lay their eggs, but because it could be a good place to mate. An argument against this interpretation is that mated females have reduced sexual receptivity compared to virgin females and that virgin females are not attracted to egg-laying sites marked with pheromones[18]. It is thus unlikely that the primary driver of mated female laying eggs on substrate with intermediate pheromone concentration is to obtain mates. Although it relies on the same pheromones, our findings suggest a different sensory pathway is involved in oviposition site selection than in mating. In the case of oviposition site selection, it is the oxidation product of 7-T-heptanal- detected through olfaction that attracts females to oviposition sites in our study. This illustrates how different sensory modalities detecting the same cues shape context-dependent behavioral responses[46,48].

Using heptanal as a cue for oviposition site selection requires that the environment breaks down the contact pheromone 7-T into a behaviorally active volatile (Fig. 3A, B). The oxidation of cuticular hydrocarbons at their double bonds regulates social interactions in several insect species[76,77]. For fruit flies, in particular, the female-specific hydrocarbon 7,11-HD autoxidizes into the aldehyde (Z)-4-undecenal, which attracts conspecifics over long distances[57]. While we found no effect of 7,11-HD on oviposition site selection (Fig. 2C), the principles of 7,11-HD and 7-T becoming behaviorally relevant after oxidation share similarities. Considering that the chemical properties of (Z)−4-undecenal and heptanal make them more volatile than the long-chain unsaturated cuticular hydrocarbons they are derived from, we are left with the possibility that the function of oxidation products of other unsaturated cuticular hydrocarbons associated with fruit flies are yet to be discovered. That females can use different doses of volatiles released by deposited pheromones to select an oviposition site, rather than through localized assessment of micro indicators of site quality[46,47,52], suggests that females may be able to gain information on the social quality of sites from a distance before choosing a communal oviposition site. Moreover, these cues remain detectable after the other females have left the site, providing lasting information about the number of prior visiting females. Additional social cues—from adult flies, eggs and larvae—could be acquired through visual and gustatory stimuli, but these were excluded in our assay. We also do not exclude that additional chemicals produced by flies may influence choices between oviposition sites[31,38]. A candidate is methyl laurate, which has an aggregative effect[44,78]. However, as we found no attraction to females that mated with Oe⁻ males (Fig. 2B; Fig. S1B; Tables S4–S5), which transfer both cVA and methyl laurate[54], we postulate that methyl laurate was unlikely to have an independent effect on the selection of oviposition sites in our assay.

### The neuronal basis of aggregation

The inverted U-shaped behavioral dose response curve to cVA and 7-T is unlikely to be explained by sensing from single odorant receptors neurons, as we found sigmoidal dose response curves by all olfactory receptors we tested (Fig. 5C). Instead, our results suggest that several neuronal networks work in parallel to integrate the olfactory cues to eventually shape the inverted U-shaped dose-response curve observed for the selection of oviposition sites. The first neuronal network consists of Or67d and Or65a ORNs and is responsible for the behavioral responses to cVA at different concentrations. Our results suggest that Or67d is involved in the attraction at low pheromone concentration, whereas Or65a seems necessary for mediating the response to higher

pheromone concentrations. Integration of these two receptors would allow for the generation of an inverted U-shaped dose-response curve (Fig. 1). Following the previous assumption that the two ORN types form a neuronal circuit in which Or65a inhibits the action of Or67d[48,67,68], our results suggest that Or67d has a higher sensitivity to cVA and allows for attraction at low concentrations of cVA (Fig. 4C). Interestingly, a recent study suggested that support cells of the Or67d ORN are necessary for sensing number of flies in their group[79], illustrating that cVA and its receptors can modulate social density-dependent behaviors in a variety of contexts.

While our results suggest that Or65a has a lower sensitivity to and mediates attraction to higher levels of cVA (Fig. 4C), there is a controversy concerning the detection of cVA by Or65a ORNs. While van der Goes van Naters and Carlson (2007)[80] report a weak sensitivity of Or65a to cVA, Pitts et al.[81], and Wu et al.[82] found no physiological evidence for the detection of cVA by Or65a ORNs. Pitts et al.[81] did however find a functional connection between behavior normally attributed to cVA and the optogenetic stimulation of Or65a ORNs, leading them to conclude that Or65a neurons are activated by some unknown compound related to cVA. It is possible that Or65a detects an oxidation product rather than cVA itself. This brings us back to our suggestion that there are yet to be discovered oxidation products associated with fruit fly pheromones that mediate behavioral responses.

When it comes to selection of oviposition sites based on pheromone dose, cVA by itself is not sufficient. We suggest that the second neuronal network required senses 7-T's oxidation product heptanal, which serves as a dose-independent activator necessary for the dose-dependent response to cVA. Considering that at least five different olfactory receptors respond to heptanal (Or13a, Or22a, Or35a, Or67b, and Or69a), we assume that the signals from these receptors need to be integrated with that from cVA in a higher brain region receiving a multitude of olfactory inputs. This awaits further investigation but a prime candidate would be the lateral horn given that it is the site of olfactory integration[83,84]. In support of this hypothesis, it has recently been shown that female-specific neurons in an anterior dorsal neuron cluster in the higher brain regions close to the lateral horn are necessary for communal oviposition in response to *Drosophila* pheromones[85].

Given our finding that food conditions modulate the response to pheromones (Fig. 1C, D), the third neuronal network involved in group joining behavior would need to assess the food conditions of the oviposition site. This is in line with the finding that the resource condition of a substrate directly affects the number of individuals it can sustain[86–88]. Odors indicating the presence of larval resources at an oviposition site generally include acetic acid and other yeast fermentation products[89,90]. These odors are generally detected by the ionotropic receptors, rather than the olfactory receptors that respond to the pheromones[18,91,92]. Interestingly, heptanal has a pervasive fruit- to butter-like smell and is found in the headspace of fruits and yeast[93–95]. Furthermore, we found that Or22a ORNs had the highest affinity to heptanal (Fig. 5), which is a receptor responding to food odors[96–98]. Considering that Or22a ORN is necessary for the response to 7-T (Fig. 5), it is conceivable that heptanal is perceived as a food cue rather than strictly a social cue. This is intriguing as a previous work by Duménil et al.[18] showed that flies evaluate communal oviposition sites based on the presence of social cues and food, leading to the possibility that heptanal might work as a qualitative cue for both. Given that heptanal does not work in a dose-dependent manner, the effect of heptanal will not be lost due to the natural presence of heptanal in the headspace of fruits and yeast[93–95]. The overlap between the sensing of food odors and pheromones produced by males would offer an explanation to several studies that have shown that cVA only holds a behavioral relevance for oviposition when it is sensed in the presence of fruits and fermentation products[20,41,99]. The need for this synergistic effect solves the issue of the many functions cVA has on the behavior of *Drosophila*, like inducing aggregation for mating[41], acting as an aggression inducing cue for males[67], or acting as an aphrodisiac for females or anti-aphrodisiac for males[54,100]. Considering that neither cVA nor any of the cuticular hydrocarbons are able to instigate oviposition by themselves (Fig. 2C), our results lead to the intriguing possibility that flies circumvent the poor attractiveness of substrates with low nutritional content by mimicking food odors that synergistically work with cVA for oviposition site selection. This suggests that pheromone cues relevant for oviposition site selection might be using a neuronal network common to those for sensing food[83,101].

As almost all animal species form groups during parts of their life cycles[102,103], it is important to understand the mechanisms underlying aggregation. The specific cues leading to group formation and the decisions that individuals make, as well as their mode of action, are relevant to understand given the declining populations of insect species[104], the expanding distributions of pests[105], and the need to control disease vectoring insects[106]. This is because group formation is crucially important for fitness[8,103], and is therefore expected to be under strong selection for optimal decision making. While we are still at an early stage in understanding how the integration of different pheromones translates to quantitative behavioral responses, our study exemplified that olfactory cues can lead to responses that are consistent with expectations of whether joining a communal oviposition site would have a positive or negative effect on fitness. More specifically, we found that flies can adjust their behavior according to the pheromone concentrations and resource conditions through a combination of olfactory cues that attract at intermediate concentrations and inhibit it at high concentrations. These findings may open new areas of interest in determining how the brain determines social group size, and may find its relevance in the prediction on how insects aggregate in nature and for the development of sustainable pest-control strategies.

## Methods
### *Drosophila* rearing and stocks
All flies were reared under controlled conditions (25 °C, 50% RH, 12:12 L:D) in 170 mL rearing bottles on a food medium containing agar (10 g/L), cornmeal (15 g/L), glucose (30 g/L), molasses (30 g/L), propionic acid (5 mL/L), soy flour (10 g/L), sucrose (15 g/L), tegosept (10 mL/L), wheat germ (10 g/L) and yeast (35 g/L). Virgin flies were separated by sex under $CO_2$ anesthesia on the day of eclosion and aged for five to seven days in groups of 10 in 25 × 95 mm rearing vials. All behavioral experiments were conducted with $w^{1118}$ [BDSC# 6326], unless otherwise specified. Neuronal activity-silenced flies were generated by crossing Orco-Gal4 [BDSC# 23292], Or13a-Gal4 [BDSC# 9945][107], Or22a-Gal4 [BDSC# 9951][64], Or35a-Gal4 [BDSC# 9968][107], Or65a-Gal4 [BDSC# 9993][108], Or67b-Gal4 [BDSC# 9995][107], Or67d-Gal4 [BDSC# 9997][108], Or69a-Gal4 [BDSC# 10000][108] and Gr32a-Gal4[58] to UAS-Kir2.1 [BDSC# 6595][109]. Oenocyte-ablated (Oe⁻) flies were generated by crossing +;PromE(800)-Gal4, tub-Gal80ᵗˢ;+ with +;UAS-StingerII, UAS-hid/CyO;+[56]. Control flies were generated by crossing +;PromE(800)-Gal4, tub-Gal80ᵗˢ;+ with +;UAS-StingerII;+[56]. Oe⁻ and control strains developed at 18 °C until eclosion, after which they were transferred to 25 °C for seven days prior to ablate the oenocytes prior to the pheromone extraction. Flies with calcium indicators were generated by crossing Or13a-Gal4 [BDSC# 9945], Or22a-Gal4[96], Or35a-Gal4 [BDSC# 9968], Or47b-Gal4 [BDSC# 9984][64], Or67b-Gal4 [BDSC# 9995] and Or69a-Gal4 [BDSC# 10000] to UAS-GCaMP7b [BDSC# 79029].

### Oviposition experiments
The oviposition experiment follows the protocol reported in Verschut et al.[110]. Five-to-seven-day old flies were mated in groups of 10 males by 10 females for three hours in rearing vials containing food. Afterwards, females were individually kept for 18 hours in 2 mL screw cap vials

(72.609/65.716 · Sarstedt) containing an agar medium of 300 µL 3% bacteriological agar in the bottom (CAS: 9002-18-0 · Becton Dickinson Difco) and 150 µL yeast extract paste as a food source on the inside of the caps (CAS: 8013-01-02 · Fisher BioReagents). At the start of the experiment, females were individually transferred into disposable two-choice oviposition assays consisting of 57 × 38 × 17 mm (L × W × H) rectangular polystyrene dishes (31170 · Bodemschat, the Netherlands). The assays contained two 10 × 38 × 4 mm oviposition zones of 0.75% agar containing 100 mM of D-Sucrose (99.7% · CAS: 57-50-1 · Acros Organics) at either ends of the assay, that were separated by a 3% agar middle zone of 37 × 38 × 4 mm of that was unsuitable for oviposition[111]. The effect of food availability was tested by including 100 mM of D-Sucrose and 8.75 g/L yeast in the 0.75% agar oviposition zones. This quantity of yeast represents 25% of the yeast content of the food medium on which the flies developed. Each oviposition zone enclosed a 0.2 mL PCR tube cap (72.737.002 · Sarstedt) containing 3 mm filter paper discs (Grade 1 Chr − 0.18 mm thick · Whatman), loaded with either 10 µL of the predetermined pheromone concentration diluted in n-Hexane or n-Hexane as solvent control (≥99% · CAS: 110-54-3 · Acros Organics). The solvent was swiftly evaporated under a nitrogen flow and the cups were covered with a fine polyamide mesh preventing physical contact between the fly and the odor treatments (see Fig. S1D for solvent control treatments).

The quantitative chemical cue was manipulated by loading increasing doses of pheromone extracts of individual flies onto the filter paper discs (10 µL per individual−see Pheromone Extraction). The pheromones and the n-Hexane solvent control were loaded in steps of 10−30 µL until the desired group size was reached. At each step the n-Hexane was directly evaporated under the nitrogen flow to reduce any potential effects of solvent build-up. By using pheromone extracts, we ensured that resident adult flies could not inoculate the substrates with microorganisms and cause an uncontrolled bias between the two oviposition zones. As extracts may over-represent the concentration of pheromones a fly can deposit on a substrate, several smaller fractions of the pheromone extract were tested. Pheromone equivalents of cis−11-Vaccenyl Acetate (>98%− CAS: 6186-98-7−Pherobank, the Netherlands), 7-Tricosene (CAS: 52078-42-9−see Synthesis of 7-Tricosene), 9-Tricosene (96%−CAS: 27519-02-4 −Alfa Aesar) and 7,11-Heptacosadiene (CAS: 100462-58-6−(Billeter et al.[59])) representing two, six and twelve mated females were based on concentrations reported by Laturney and Billeter[54]. For the 7-T oxidation products, heptanal (97%− CAS: 111-71-7−Alfa Aesar), heptanoic acid (≥99%−CAS: 111-14-8−Sigma Aldrich), hexadecanal (≥98%−CAS: 629-80-1−Cayman Chemical) and palmitic acid (≥99%−CAS: 57-10-3−Sigma Aldrich), we divided the molecular weight of 7-T with that of each oxidation product to calculate a hypothetical conversion factor in which all 7-T molecules would break down into that specific compound. Odor saturation was prevented in the assays by covering them with Parafilm that was punctured several times with a fine needle above the oviposition zones for air circulation. All experiments were run in a 140 × 47 × 77 cm (L × W × H) enclosure made of aluminum profiles (T-slot 30 × 30 mm N-O/BSB− Techniek Specialist, the Netherlands) and ABS plates (8 mm black−S-Polytec GmbH, Germany). The enclosure was placed under controlled conditions (25 °C, 50% RH) and effectively eliminated external disturbance that may affect the oviposition choice. A 12:12 L:D cycle was maintained using 4000−5000 K white LED strips (3528−IP65 60 LED/M 12 Volt−RoHS) and 650−660 nm deep red LED strips (3528−IP65 60 LED/M 12 Volt−RoHS). The light was diffused through a combination of a 27% light transmission (3 mm PETG AR030−Pyrasied acrylic, the Netherlands) and a 38% light transmission diffusor plate (3 mm PETG AR050−Pyrasied acrylic, the Netherlands) covering the LED lights at a distance of 15 cm. Odor saturation was prevented by 120 mm axial fans that pushed and pulled fresh air through the enclosure at all times (F12 PWM− Artic GmbH, Germany). After 24 h the number of eggs laid in each oviposition zone was

counted under a stereomicroscope and the oviposition indices were calculated as follows: (Eggs side A − Eggs side B) / (Eggs side A + Eggs side B).

## Pheromone extraction

Pheromones were washed from the cuticle of seven-day-old w[1118], oenocyte-ablated or oenocyte-control flies of the indicated sex or mating state with n-Hexane (≥99%−CAS: 110-54-3−Acros Organics). All flies were collected under $CO_2$ anesthesia on the day of eclosion, separated by sex, and aged in groups of 10 in rearing vials. Mated females were acquired as previously described for the oviposition experiment. Prior to the extraction the flies were anesthetized on ice and groups of 80 flies were transferred into 2 mL glass screw cap vials (29378-U−Supelco). We added 12 µL of n-Hexane per fly, vortexed them for 3 min, and transferred the supernatant into a clean 2 mL glass vial. Due to minor evaporation and absorption of the n-Hexane by the fly bodies, the resulting supernatant contained ∼10 µL of pheromone extract per fly, which was the dose used to represent a single fly in the behavioral experiments.

## Pheromone quantification

To quantify the concentration of pheromones deposited by groups of flies, we kept one, two, six and 12 five-day-old w[1118] males in 75 × 12 × 1 mm glass tubes (47729-570−VWR) for 90 min (n = 8−10 per group size). The tubes were closed with a cotton plug, leaving ∼1 mL of volume for the flies to interact in. The vials were placed upside down to ensure that the flies would walk on the glass rather than the cotton plug. After 90 min, the flies were removed and 1 mL of n-Hexane (≥99% −CAS: 110-54-3−Acros Organics) was pipetted into each empty tube and swirled by hand to extract the deposited pheromones. Afterwards, the liquid was pipetted into 2 mL glass screw cap vials (29378-U− Supelco) and the n-Hexane was evaporated under a nitrogen flow to only leave the extract of deposited pheromones. The extract was resuspended with 40 µL of standard solution, consisting of 10 ng/uL Octodecane (99%−CAS: 593-45-3−Sigma-Aldrich) + 10 ng/µL Hexacosane (99%−CAS: 630-01-3· Sigma-Aldrich) dissolved in n-Hexane, vortexed for 2 min and pipetted in a new glass vial in preparation of the analysis by Gas Chromatography coupled with Flame Ionization Detection (GC-FID) (volume of injection 5 µL). In addition, we transferred a single fly from each of the groups into 2 mL glass screw cap vials (29378-U−Supelco) to extract the pheromones present on the cuticle of that single fly (n = 7−9 per group size). The pheromones were extracted by pipetting 50 µL of standard solution and vortexing the vial for 2 min at minimum speed. Afterwards, the flies were removed and the extract was analyzed by GC-FID (volume of injection 2 µL). The GC-FID analysis was done on an Agilent 7890 Gas Chromatograph coupled with an DB-1 column (20 m × 0.18 mm × 0.18 µm; Agilent Technologies, USA) Flame Ionization Detector, using a splitless injector set at 250 °C with helium as carrier gas (flow: 37.2 cm/s⁻¹). The column oven temperature was programmed from 50 °C, for 1.5 min, to 150 °C, at 10 °C min⁻¹, then to 280 °C at 4 °C min⁻¹, and held for 5 min[18]. ChemStation software (Agilent technologies) was used to integrate compounds based on peak areas relative to the internal standard, as described in Krupp et al.[112].

## Transcuticle calcium imaging

The transcuticle calcium imaging was performed as described by Vulpe et al.[70] on six-day-old females that were mated 24 h prior to the experiment. These mated females were wedged into the narrow end of a 200 µL truncated plastic pipette tip to expose the antenna. The antennae were stabilized between a tapered glass microcapillary and coverslip covered with double-sided tape. We loaded 4.5 µL pure 7-Tricosene (CAS: 52078-42-9−Cayman Chemicals) or 100 µL 10⁻⁵ till 10⁻¹ aliquots of heptanal (97%−CAS: 111-71-7−Alfa Aesar) in paraffin oil (CAS: 8012-95-1−Sigma Aldrich) onto filter paper discs as odor

treatments. The odor pulses were controlled through pCLAMP 10.4 (Molecular Devices) and delivered from a close range to the antenna via a 500-ms air pulse at 250 mL/min at close range (for 7-T[113]) or at 200 mL/min at long range through the main airstream of 2000 mL min (for heptanal). The calcium imaging was performed on a BX51WI Olympus microscope equipped with a scientific CMOS camera (Prime 95B, Photometrics) and a Universal LED Illumination System (pE-4000, CoolLED). The images were acquired using Micro-Manager software (UCSF). Motion correction was performed by StackReg (ImageJ plugin) and the Regions of Interest (ROIs) were identified from cells responding to diagnostic odorants applied as positive controls and selected using a custom Python code based on PyCharm (JetBrains). The maximum ΔF/F for each ROI was determined using Fiji (NIH) and sample traces were obtained by averaging individual traces and smoothed using a binomial filter with Igor Pro 6.3 software (Wave-Metrics). The recordings were made on six to seven individuals per genotype and the calcium responses of three ROIs were averaged per recording to represent the antennal response of that individual.

### Synthesis of (Z)−7-Tricosene

(Z)−7-Tricosene was synthesized following the protocol of Billeter et al.[56]. A stirred suspension of (1-hexadecyl)triphenylphosphonium bromide (CAS: 14866-43-4−Alfa Aesar−9.3 g, 16.40 mmol, 1.07 eq) in dry THF (88 mL) was treated with a solution of sodium hexamethyldisilazane (CAS: 1070-89-9−Sigma-Aldrich−16.4 mL, 16.40 mmol, 1 M in THF, 1.07 eq) through dropwise addition at 0 °C. The resulting orange-red reaction mixture was warmed up to room temperature and stirred for 1 h. The reaction mixture was cooled down to −40 °C and heptanal was added dropwise (CAS: 111-71-7−Sigma-Aldrich−1.75 g, 15.33 mmol). The reaction was warmed up to room temperature overnight and the next day the reaction was quenched with water (70 mL), and extracted with pentane (CAS: 109-66-0−Honeywell−3 × 100 mL). The combined organic layers were washed with brine (150 mL), dried over MgSO$_4$ and concentrated in vacuo. The crude product was purified by column chromatography using pure pentane, yielding a colorless oil. Next, the purified product was recrystallized from acetone (25 mL) at −20 °C, yielding (Z)−7-Tricosene (7-T) (3.95 g, 12.24 mmol, 80%).

$^1$H- and $^{13}$C-NMR spectra were recorded on an Agilent MR400 (400 and 100.59 MHz, respectively). CDCl$_3$ was used as solvent unless stated otherwise. Chemical shift values are reported in ppm with the solvent resonance as the internal standard (CDCl3: δ7.26 for $^1$H, δ77.16 for $^{13}$C). Data are reported as follows chemical shifts, multiplicity (s = singlet, d = doublet, dd = double doublet, ddd = double double doublet, dt = double triplte, td = triple doublet, t = triplet, q = quartet, p = pentet, b = broad, m = multiplet), coupling constants J (Hz), and integration. $^1$H NMR (400 MHz, CDCl$_3$) δ 5.42−5.27 (m, 2H), 2.01 (overlapped dt, J = 6.4 Hz, 4H), 1.47−1.13 (m, 34H), 0.88 (td, J = 7.0, 6.0, 3.6 Hz, 6H). $^{13}$C NMR (101 MHz, CDCl$_3$) δ 130.07, 130.05, 32.09, 31.95, 29.94, 29.91, 29.86 (5C), 29.82 (2C), 29.72, 29.53, 29.48, 29.16, 27.38, 27.37, 22.85, 22.82, 14.27, 14.26. All reactions were performed using flame-dried glassware under an atmosphere of nitrogen by standard Schlenk techniques and dry solvents. Reaction temperatures below 0 °C refer to internal temperatures, while reaction temperatures higher than room temperature refer to heating bath temperatures. Dry solvents were taken from a MBraun solvent purification system (SPS-800). TLC analysis was performed with Merck silica gel 60/Kieselguhr F245, 0.25 mm. Compounds were visualized using either anisaldehyde stain, EtOH (135 ml), H$_2$SO$_4$ (5 ml), AcOH (1.5 ml), p-anisaldehyde (CAS: 123-11-5 - TCI−3.7 ml), or elemental iodine. Flash chromatography was performed using SiliCycle silica gel type SiliaFlash P60 (230–400 mesh) as obtained from screening devices.

### Oxidation of 7-Tricosene

The production of heptanal as an oxidation product of 7-Tricosene was analyzed by pipetting 10 µL of synthetic 7-T into a 20 mL headspace vial closed with a screw cap with septum (5188-2753/5188-2759−Agilent Technologies). We performed a measurement directly after pipetting 7-T into the vial (i.e. 0 h) and after an oxidation period of 24 h under ambient room temperature. The resulting headspace in the 20 mL headspace vials were measured on a Gas Chromatograph Mass Spectrometer (GCMS-QP2010−Shimadzu) equipped with a nonpolar HP5-MS column (0.25 µm, 0.25 mm × 30 m−Agilent Technologies) and a headspace autoinjector (AOC-5000−Shimadzu) with the agitator temperature set a 120 °C, the syringe temperature at 120 °C and the agitation time at two minutes. The injector temperature was programmed at 225 °C with split ratio at 10 and column flow at 1. The column temperature was set at 30 °C and increased 8 °C per minute until 225 °C was reached and maintained for 5 min. The electron impact ionization was set at a selective ion monitoring of m/z 70 to improve the detection of heptanal. The identity of heptanal peaks were validated using synthetic heptanal (≥95%−CAS: 111-71-7−Sigma Aldrich) diluted in n-Heptane (99%−CAS: 142-82-5−Biosolve).

### Statistical analysis

The oviposition indices were first analyzed per treatment with a two-tailed Wilcoxon Signed Rank tests with mu = 0 assuming no preference to either of the treatments. Preferences across pheromone equivalents or genotypes were compared using a generalized linear model (GLM), or a generalized additive model included a smoothing term (GAM), with a quasibinomial error distribution to account for overdispersion. We used a model comparison to determine whether the GAM outcompeted the GLM based on a change in degrees of freedom and deviance, effectively showing whether the behavioral response is linear or non-linear. In order to use the quasibinomial error distribution, the data was analyzed using a 'cbind' including the number of eggs laid on either side of the assay. This generated values between 0 and 1 representing aversion or attraction to the pheromone treatment respectively. Where applicable, the distribution of the GLM and GAM were used to derive a model fit and confidence intervals that allowed the visualization of the data as linear or non-linear distributions. The model assumptions were checked by estimation of overdispersion and inspections of model residuals. The concentrations of pheromones on the cuticle of individual $w^{1118}$ males and the pheromones deposited by groups of $w^{1118}$ males were compared and analyzed per treatment using GLMs. The concentrations of heptanal at 0 and 24 hours of oxidation were compared with a Kruskal−Wallis one-way analysis of variance to account for the non-parametric data. The trans-cuticle calcium imaging results were compared with two-tailed t-tests comparing the responses to the solvent with that of either pure 7-T or 10$^{-1}$ heptanal. The analyses were carried out in R (v. 3.6.1; R Foundation for Statistical Computing, Vienna, AT). The GLMs were performed using lme4[114], the GAMs with mgcv[115], and car[116] for model comparisons. Tukey-HSD tests were performed with multcomp[117]. The data was visualized using ggplot2[118].

### Reporting summary

Further information on research design is available in the Nature Portfolio Reporting Summary linked to this article.

## Data availability

The data sets generated during the current study are available in the DataverseNL repository: https://doi.org/10.34894/YGEFCT. Source data are also provided with this paper. Source data are provided with this paper.

## Code availability

The script used for the current study is available in the DataverseNL repository: https://doi.org/10.34894/YGEFCT.

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

## Acknowledgements

The authors thank I. Pen for his advice on statistical analysis. This research was supported by the Swedish Research Council Vetenskapsrådet (grant VR-2018-0354 to T.A.V.), the Dr. J.L. Dobberke Foundation for Comparative Psychology (to T.A.V.), the Dutch organization for scientific research (grant ALWOP.611 to J.C.B.) and the National Institutes of Health (grants R01DC016466, R21DC018912, and R21AI169343 to C.Y.S.).

## Author contributions

Project design: T.A.V., M.A.C., B.W., and J.C.B. Behavioral experiments: T.A.V. 7-T synthesis and oxidation: G.L.C., R.J.L.S., and A.J.M. Pheromone deposits: N.P.D. and J.C.B. Transcuticle Calcium Imaging: R.N. and C.Y.S. Data analysis and visualization: T.A.V. Illustrations: T.A.V. Writing: T.A.V. and J.C.B. with input from all authors.

## Competing interests

The authors declare no competing interests.
