## [Peer Review File · Nature Communications]

Aggregation pheromones have a non-linear effect on oviposition behavior in *Drosophila melanogaster*Reviewers' Comments:

Reviewer #1:

Remarks to the Author:

This study employs a sophisticated set of tools including two-choice behavioral bioassays, chemical analysis, *Drosophila* genetics, and optophysiology to determine how females use volatile cues to estimate group size in evaluating suitable oviposition sites. The authors suggest that cuticular extracts of both males and mated females impact oviposition choice, and that this is mediated by the olfactory perception of both *cis*-vaccenyl acetate and heptanal, an oxidation product of the male component 7-T. Using the Gal4 system to selectively silence ORNs and transcuticular calcium imaging of the antenna, they show that Or67d and Or65a both impact oviposition choice through the detection of cVA, and several receptors seem to be utilized in the detection of heptanal.

The experiments are competently performed, and the data appear sound. However, while I do not have concerns with the results, I have two major concerns with their interpretation and discussion. I outline these below:

CONTEXT

In general, the use of chemical cues to assess oviposition sites is a well-studied issue across the invertebrate phyla - both aquatic and terrestrial. Further, the use of conspecific cues in relation to these sites (here referred to as group size cues) has also been looked at in many systems, especially insects like Lepidopterans, but also some benthic invertebrates as well. While reviewing all this literature would be tedious, it would be good that the authors refer to the existence of such knowledge, with some examples, as the phenomenon is not unique to *Drosophila*.

ASSUMPTIONS

1. Group Size: Throughout the manuscript, the authors insist that *Drosophila* exhibit group-size dependent oviposition. But, they never actually assess group size in this study. Groups are never themselves tested, nor altered. Rather, they use the summation of extracts from individuals as a proxy for groups. However, this makes the assumption 1) that the flies would perceive a group of flies in a dose-dependent manner equivalent to the extracts and chemicals used, and 2) that the chemicals actually are a proxy for groups at all, which is never assessed in this study. To be clear, in Ln 89 it is noted "We surmise that the accumulation of pheromonal deposits may inform females about the number of flies that have visited that communal oviposition site.". Yet, to my understanding, this was never actually tested, but is assumed for the rest of the study. It is possible that cVA acts as an oviposition deterrent, which has nothing to do with group size itself, but rather the previous presence of individuals, as has been shown in many other systems. This would also explain the monotonic decrease in oviposition preference with increasing extract equivalents in the presence of a suitable substrate (sugar and yeast).

2. Substrate Quality: The authors also assume that the flies are judging substrate quality (this stems from a comparison between a sugar only and sugar+yeast substrate). But again, substrate quality is never directly assessed beyond the dichotomous presentation. Rather the comparison to quality should have been made only after showing that the oviposition choice changes according to substrate quality itself, and then compare this to the impacts on chemical presence.

3. Heptanal: The authors show that heptanal is an oxidation product of 7-T. Then they show that heptanal impacts oviposition. But, this assumes that the flies are detecting heptanal as a product of 7-T and not as a plant volatile (a point the authors do note in the discussion). In fact, heptanal is a prominent component of citrus fruit, which is the ancestral host of *Drosophila*. It seems more likely that the flies are simply detecting it as a fruit volatile. One could assess this by comparing the typical volatile concentration of the 7T oxidation product vs. a fruit. If the amount of heptanal is much greater in the fruit, then it would surely drown out the signal from 7T products.

4. Density-dependence: On In 327, the authors note "dose-dependent cue indicating whether communally laying eggs may result in positive- or negative density-dependent effects". As with point #1, this cannot be stated when density-dependence is itself not assessed.

5. Role of volatile compounds: On In 346, the authors note that "females can gain information on the social quality of sites from a distance before choosing a communal oviposition site". This assumes that the flies can detect these very heavy volatiles (except heptanal) from a distance, which is not very likely beyond the substrate itself.

6. Statistics: I am not a statistical expert, but I am concerned with the interpretation of a couple of experiments, particularly in Figure 1 and Figure 5. While I see that the authors used "a generalized linear model (GLM), or a generalized additive model included a smoothing term (GAM), with a quasibinomial error distribution to account for overdispersion", to the naked eye the data in Figure 1 is highly overlapping for all points and it is not clear that there is any substantial difference - usually for such simple experiments like two-choice behavioral assays, the statistical differences should be more apparent when graphically represented. Contrary to this, in Figure 5 the authors note In 309 "Considering that this response is barely above the background signal, we assume that this statistical difference is unlikely to hold biological relevance and that 7-T is unlikely to be detected by this receptor.". It seems a bit strange to accept statistical relevance in some cases and not others based on assumptions.

OTHER MINOR ISSUES

1. It is not clear why male-produced chemicals are influencing oviposition at all. I know the authors reference this in the discussion, but it seems rather counter intuitive. Could the authors spend more time explaining this?

2. In 298 "silencing neurons expressing these ORs abolished the preference towards these pheromones". Abolished is a rather strong word that is not represented in the data. Also in this section, the statement that the Ors work in concert for perception cannot be assessed from this study, only that they all detect heptanal.

3. In 358 "fitness benefits of laying eggs communally ultimately come from the number of larvae developing at the communal site". Yet the preference decreases with increasing concentrations of fly-produced chemicals in this study!

4. It's not clear from the methods if when adding increasing equivalents of individuals to the filter paper more solvent was used as well. If so, this could confound the results, as increased solvent buildup will not evaporate in the same way. Can the authors clarify this?

5. The control experiment indices for the experiments in figures 1,2, and 3 are missing. What happens with no treatment?

Reviewer #2:

Remarks to the Author:

Animals use pheromones to communicate various types of information, many of which are likely still uncovered. Verschut et al. investigated whether the full-body pheromone extracts of *Drosophila melanogaster* communicate information about the attractiveness of the deposited sites. As hypothesized, they found that the pheromone extracts attracted conspecifics following an inverted U-shaped dose-response relationship. They then identified cVA and heptanal, an oxidated derivative of 7-Tricosene, to be major components of the extract responsible for the behavioral effect. It was cVA but not heptanal that determined the strength of attractiveness. Finally, they identified olfactory

receptor neurons mediating neural and behavioral responses of these compounds. Characterization of the nonlinear dose-response relationship between pheromone extracts and the attractive behavior as well as the discovery of the function of oxidated product of a pheromone represent a solid advancement in the field. The experiments and analyses are thoroughly conducted, and the results are clearly presented. I, however, have one major issue before supporting the study.

Major comments

1. The authors describe that they have found how the resident group size modulates oviposition site selection, but this expression is too strong. Below is just a partial list of expression that mislead me.

Title : Resident group size modulates oviposition site selection in *Drosophila melanogaster*

Abstract

Line 37: We show that female fruit flies adjust their oviposition site selection to favor groups that are neither too small nor too large.

Line 40: While the dose of cVA indicates group-size,

Summary paragraph at the end of Introduction

Line 118: The determination of group size depends on the involvement of Or67d and Or65a,
Results

Line 125: Section title: Group size-dependent selection of communal oviposition sites

None of the experiments directly examined the effect of group size on the females' perceptual decisions. As many more factors besides pheromone concentration are known to come into play at the site occupied by the actual group such as visual and auditory input of others, physical interaction between animals, release of additional stress pheromones etc., one cannot replace the amount of pheromone extracts collected at one point with group size. The importance of the study does not fade at all by describing the fact more faithfully, and in fact, would rather increase. I formed a very different expectation about the study at the beginning as the term "group size" already appears in the title and the abstract. This term should be used to place the motivation and finding into a big context only in part of the introduction and discussion.

Minor comments

2. Line 92: "We also hypothesized that group size indicators may be volatile, giving individuals the opportunity to assess the group size of several sites before joining."

Line 105: "We also expect that this decision should be made prior to joining the group."

The term join implies that once a fly decides to go into the group, it will remain as a member for an extended period. Has this been systematically investigated? Please provide the literature describing the behavior.

3. Line 273: "These results suggest that Or65a and Or67d may determine group size at different doses of cVA, and that they together determine the shape of the dose-response curve found for increasing doses of pheromone extracts."

This can be relatively easily examined experimentally by expressing Kir using a combination of two Gal4 lines.

4. Line 355: "The requirement for the simultaneous sensing of these two pheromones prevents mated females from joining groups consisting of virgin females."

There is no evidence that sensing of two pheromones "prevents" attraction.

5. Line 388: "whereas Or65a seems necessary for the inhibition when groups become too large."

This does not seem to match the results shown in Fig. 4C.

6. Line 430: "Considering that Or22a ORN is necessary for the response to 7-T (Fig. 5), it is conceivable that heptanal is perceived as a food cue rather than strictly a social cue in the context of group size determination."

Given this, there is a possibility that food odor-sensitive ORNs that are not examined in this study are also involved in the behavior examined in the study. It might be worth discussing this issue.

7. Fig. 2B: It was difficult to interpret the labels. If the four labels at the bottom describe the genotype of the depositor, what do Oe- (light purple) and w[1118] (dark purple) above those labels mean?

DETAILED ANSWERS TO REVIEWERS' COMMENTS

REVIEWER #1 (Remarks to the Author):

This study employs a sophisticated set of tools including two-choice behavioral bioassays, chemical analysis, *Drosophila* genetics, and optophysiology to determine how females use volatile cues to estimate group size in evaluating suitable oviposition sites. The authors suggest that cuticular extracts of both males and mated females impact oviposition choice, and that this is mediated by the olfactory perception of both *cis*-vaccenyl acetate and heptanal, an oxidation product of the male component 7-T. Using the Gal4 system to selectively silence ORNs and transcuticular calcium imaging of the antenna, they show that Or67d and Or65a both impact oviposition choice through the detection of cVA, and several receptors seem to be utilized in the detection of heptanal.

The experiments are competently performed, and the data appear sound. However, while I do not have concerns with the results, I have two major concerns with their interpretation and discussion. I outline these below:

CONTEXT

Comment 1.1: In general, the use of chemical cues to assess oviposition sites is a well-studied issue across the invertebrate phyla - both aquatic and terrestrial. Further, the use of conspecific cues in relation to these sites (here referred to as group size cues) has also been looked at in many systems, especially insects like Lepidopterans, but also some benthic invertebrates as well. While reviewing all this literature would be tedious, it would be good that the authors refer to the existence of such knowledge, with some examples, as the phenomenon is not unique to *Drosophila*.

Response 1.1: As the reviewer pointed out, the use of aggregation pheromones is not unique to fruit flies and we had not done justice to the available literature on this matter. Therefore, we included a statement at the start of the second paragraph of the introduction (Page 3 / Line 85) to exemplify the function of pheromones in several behavioral processes displayed by different species.

ASSUMPTIONS

Comment 1.2: Group Size: Throughout the manuscript, the authors insist that *Drosophila* exhibit group-size dependent oviposition. But, they never actually assess group size in this study. Groups are never themselves tested, nor altered. Rather, they use the summation of extracts from individuals as a proxy for groups. However, this makes the assumption 1) that the flies would perceive a group of flies in a dose-dependent manner equivalent to the extracts and chemicals used, and 2) that the chemicals actually are a proxy for groups at all, which is never assessed in this study. To be clear, in ln 89 it is noted "We surmise that the accumulation of pheromonal deposits may inform females about the number of flies that have visited that communal oviposition site.". Yet, to my understanding, this was never actually tested, but is assumed for the rest of the study. It is possible that cVA acts as an oviposition deterrent, which has nothing to do with group size itself, but rather the previous presence of individuals, as has been shown in many other systems. This would also explain the monotonic decrease in oviposition preference with increasing extract equivalents in the presence of a suitable substrate (sugar and yeast).

Response 1.2: We are grateful to both reviewers for raising the important issue that we had not formally established that pheromones deposited by flies on a substrate act as a proxy for group size. We have now performed a new experiment (see new figure 1B) that demonstrates the relationship between group size and the concentration of pheromones deposited by that group. This experiment is reported on page 5/ Lines 136-155 and the method described on page 18-19/Line 592-616. We determined how much cVA and cuticular hydrocarbon pheromones individual flies deposit over 90 minutes on a substrate. We chose this duration because observation of flies in a natural setting indicated that more flies arrive at a site than leave it over

an hour, suggesting that flies tend to stay more than an hour per site (Dukas, 2021). Documenting that flies deposit pheromones in quantities that linearly correlate with resident group size now allows us to conclude that a focal female can transform a linear increase in pheromone concentrations (which we show is a proxy for resident group size) into an inverted U-shape curve attraction to an egg-laying site. This inverted U-shape curve matches both theoretical and experimental fitness benefits of joining groups of various sizes. We have updated the abstract to feature this new experiment and conclusion (see annotated abstract with track changes) that was made possible by the reviewers' queries.

Comment 1.3: Substrate Quality: The authors also assume that the flies are judging substrate quality (this stems from a comparison between a sugar only and sugar+yeast substrate). But again, substrate quality is never directly assessed beyond the dichotomous presentation. Rather the comparison to quality should have been made only after showing that the oviposition choice changes according to substrate quality itself, and then compare this to the impacts on chemical presence.

Response 1.3: We apologize for not having made clearer that the experiment suggested by the reviewer had already been published by our lab (Dumenil et al., 2016). By varying food quality and social cues in a full factorial manner, we showed earlier that females preferentially lay eggs on high nutritional quality substrate, but that this preference is modulated by social cues. In the previous paper, we did not directly manipulate CHC, but instead exposed food patches of various nutritional quality to actual flies. In the present study, we now show that the social cues – that attract females to oviposition sites containing food – are a specific combination of cVA/7-T that attract flies in an inverted U-shaped manner. We now address this more explicitly in the introduction (Page 4/Lines 111-112) and discussion (Page 12/Lines 372-384 and page 14/ lines 467-470).

Comment 1.4: Heptanal: The authors show that heptanal is an oxidation product of 7-T. Then they show that heptanal impacts oviposition. But, this assumes that the flies are detecting heptanal as a product of 7-T and not as a plant volatile (a point the authors do note in the discussion). In fact, heptanal is a prominent component of citrus fruit, which is the ancestral host of *Drosophila*. It seems more likely that the flies are simply detecting it as a fruit volatile. One could assess this by comparing the typical volatile concentration of the 7T oxidation product vs. a fruit. If the amount of heptanal is much greater in the fruit, then it would surely drown out the signal from 7T products.

Response 1.4: We thank the reviewer for asking us to better explain our view on the function of heptanal. In this manuscript, we show that the cuticular hydrocarbon 7-Tricosene is naturally degraded into the volatile molecule heptanal, and that it is detected by odorant receptors that have been previously identified as food receptors. There is thus a possibility that heptanal derived from 7-T taps into the same olfactory circuit as heptanal derived from fruit. More importantly, we now emphasize in the discussion that heptanal is not dose dependent. Instead, our data show it is needed to make cVA attractive, irrespectively of its concentration. Theoretical and functional experiment (discussed on page 11/lines 361-363) have previously indicated that an oviposition site is of good quality (ie supportive of offspring development) when it either contains sufficient nutritional resources or if it is already used by conspecifics. It is therefore likely that evaluation of the quality of a site goes through the same cue - heptanal - which can be either a food or a social cue. Given that heptanal does not work in a dose-dependent manner, the effect of heptanal will not be lost due to the natural presence of heptanal in the headspace of fruits and yeast. We have developed this point more clearly in the discussion (Page 14/lines 467-472).

Comment 1.5: Density-dependence: On ln 327, the authors note “dose-dependent cue indicating whether communally laying eggs may result in positive- or negative density-dependent effects”. As with point #1, this cannot be stated when density-dependence is itself not assessed.

Response 1.5: We thank this reviewer for this comment and agree that we had not demonstrated this in the first version of the manuscript. The experiments described in response 1.2 and shown in Figure 1b, now addresses this issue by showing that pheromones at a location increase linearly in a group size-dependent manner.

Comment 1.6: Role of volatile compounds: On ln 346, the authors note that “females can gain information on the social quality of sites from a distance before choosing a communal oviposition site”. This assumes that the flies can detect these very heavy volatiles (except heptanal) from a distance, which is not very likely beyond the substrate itself.

Response 1.6: The measurements of the two-choice oviposition assays are 57 x 38 x 17 mm (L x W x H – see figure 1A). Practically, this means that the fly can only display post-alighting search behavior as it is already in close proximity to the substrate. We expect that the flies have the necessary means to perceive the volatiles we tested, and have not made any claims that the flies can sense the olfactory cues beyond the distance of the assay. However, previous studies have shown that fruit flies can perceive and respond to cVA over distances larger than a meter (in wind tunnel experiments and field studies). This makes it likely that some of the pheromones tested in our experiments also function in pre-alighting search behavior and attraction to oviposition sites over larger distances.

Comment 1.7: Statistics: I am not a statistical expert, but I am concerned with the interpretation of a couple of experiments, particularly in Figure 1 and Figure 5. While I see that the authors used “a generalized linear model (GLM), or a generalized additive model included a smoothing term (GAM), with a quasibinomial error distribution to account for overdispersion”, to the naked eye the data in Figure 1 is highly overlapping for all points and it is not clear that there is any substantial difference - usually for such simple experiments like two-choice behavioral assays, the statistical differences should be more apparent when graphically represented. Contrary to this, in Figure 5 the authors note ln 309 “Considering that this response is barely above the background signal, we assume that this statistical difference is unlikely to hold biological relevance and that 7-T is unlikely to be detected by this receptor.”. It seems a bit strange to accept statistical relevance in some cases and not others based on assumptions.

Response 1.7: We assume that the reviewer refers to the raw data points in Figure 1 as the overlapping data points. These data points are analyzed using two-tailed Wilcoxon signed rank test and are less relevant for the interpretation of the GLM and GAM. To aid the interpretation of the two tests, we reduced the size of the raw data points. These two analyses test for the trend across all treatments rather than the dichotomous visualization of two-choice behavioral tests that the reviewer refers to. The interpretation of the GAM and GLM should instead be done by the red and blue lines and their 95% confidence intervals (grey shaded area). The full results of these analyses are included in Table S3 for further interpretation if needed (referred to in-text). The differentiation between the visualization of the two tests has been explained in further detail in the caption of Figure 1. We have also adjusted the graphs and captions of Figure 3d and Figure 4 to clarify the difference between the two analyses as done for Figure 1.

For the result in Figure 5 that the reviewer refers to, we did not intend to come across as not accepting statistical outcomes based upon assumptions. Instead, we intended to discuss why we think that the statistical difference may not be representative without diminishing the importance of statistics. In order to fix this, we have adjusted our explanation of the result in Figure 5. While the change in intensity of Ca²⁺ indicators is barely above 0 when stimulated with 7-T (0.0205 ± 0.0057), we have removed the ‘unlikely to hold biological relevance’ argument and now instead mention that ‘*we surmise that this response can only have a negligible effect on sensing 7-T during oviposition site selection. Instead, we expect that this small response to 7-T may be an effect of the rapid oxidation of 7-T into heptanal upon contact with oxygen (see Fig. 3B), or an artifact of Or69a’s dual affinity to pheromonal compounds*

and environmental semiochemicals'. Testing the dual affinity of this receptor is, however, beyond the scope of our study and we cannot fully rule out the role of it in our experiments.

MINOR ISSUES

Comment 1.8: It is not clear why male-produced chemicals are influencing oviposition at all. I know the authors reference this in the discussion, but it seems rather counter intuitive. Could the authors spend more time explaining this?

Response 1.8: We thank the reviewer for this interesting question. We have previously shown that both males and mated females, but not virgin females attract other flies to oviposition sites (Dumenil et al., 2016). We also repeated this experiment in the context of the new assay we developed for the present manuscript and show the result in Supplementary figure Fig S1A. We now explain why both males and mated females are attractors on page 12/lines 371-384. In short, mated females possess the right information for oviposition, in that they are attracted to yeast, and provide the right social benefits, in that they lay fertilized eggs. Males are also naturally attracted to high quality oviposition sites due to the available mates and resources. They are therefore also an honest signaler of oviposition site quality. Virgin females are not good indicators of egg-laying site, explaining why females-specific cues are not sufficient.

Comment 1.9: In 298 “silencing neurons expressing these ORs abolished the preference towards these pheromones”. Abolished is a rather strong word that is not represented in the data. Also in this section, the statement that the ORs work in concert for perception cannot be assessed from this study, only that they all detect heptanal.

Response 1.9: We have changed “abolished to” to “reduced,”

Comment 1.10: In 358 “fitness benefits of laying eggs communally ultimately come from the number of larvae developing at the communal site”. Yet the preference decreases with increasing concentrations of fly-produced chemicals in this study!

Response 1.10: We have now clarified this sentence, by reminding the reader that the theoretical and experimental fitness benefits of joining a group do not increase linearly. Instead, they increase up to a group size and then drop, forming a non-linear inverted U-shape curve. We have added a comment on page 12/line 379-381: “This number of larvae should not be too small, in which the larvae fail to survive due to harmful fungal growth on the substrate, and not so large that strong resource competition leads to cannibalism.”

Comment 1.11: It's not clear from the methods if when adding increasing equivalents of individuals to the filter paper more solvent was used as well. If so, this could confound the results, as increased solvent buildup will not evaporate in the same way. Can the authors clarify this?

Response 1.11: We have added a statement in the methods that the pheromones and the *n*-Hexane solvent control were loaded in steps of 10-30 μ L until the desired group size was reached. At each step the *n*-Hexane was directly evaporated under the nitrogen flow to reduce any potential effects of solvent build-up (Page 17/ Line 549). In our experience, this should be sufficient to minimize any differences in solvent build-up or differences in evaporation rates and we expect no confounding factors due to this method.

Comment 1.12: The control experiment indices for the experiments in figures 1,2, and 3 are missing. What happens with no treatment?

Response 1.12: The control experiments consist of an assay in which the flies are only offered the *n*-Hexane solvent at both sides of the assay. The outcome of this experiment has been included in Figure S1D and shows that the flies have no preference for either side of the assay

when offered only the solvent. We also tested the effect of n-Hexane on egg laying in comparison to assays with no solvent at all (indicated as 'Plain'). The results of the analysis are included in the caption of Fig. S1D. To clarify where the reader can find this information, we have now included the following statement in the captions of figure 1 till 4; 'See Fig. S1D for solvent control treatments.

REVIEWER #2 (Remarks to the Author):

Animals use pheromones to communicate various types of information, many of which are likely still uncovered. Verschut et al. investigated whether the full-body pheromone extracts of *Drosophila melanogaster* communicate information about the attractiveness of the deposited sites. As hypothesized, they found that the pheromone extracts attracted conspecifics following an inverted U-shaped dose-response relationship. They then identified cVA and heptanal, an oxidated derivative of 7-Tricosene, to be major components of the extract responsible for the behavioral effect. It was cVA but not heptanal that determined the strength of attractiveness. Finally, they identified olfactory receptor neurons mediating neural and behavioral responses of these compounds. Characterization of the nonlinear dose-response relationship between pheromone extracts and the attractive behavior as well as the discovery of the function of oxidated product of a pheromone represent a solid advancement in the field. The experiments and analyses are thoroughly conducted, and the results are clearly presented. I, however, have one major issue before supporting the study.

MAJOR COMMENTS

Comment 2.1: The authors describe that they have found how the resident group size modulates oviposition site selection, but this expression is too strong. Below is just a partial list of expression that mislead me.

- Title:** Resident group size modulates oviposition site selection in *Drosophila melanogaster*
- Abstract - Line 37:** We show that female fruit flies adjust their oviposition site selection to favor groups that are neither too small nor too large.
- Abstract - Line 40:** While the dose of cVA indicates group-size.
- Introduction - Line 118:** The determination of group size depends on the involvement of Or67d and Or65a.
- Results - Line 125:** Section title: Group size-dependent selection of communal oviposition sites

None of the experiments directly examined the effect of group size on the females' perceptual decisions. As many more factors besides pheromone concentration are known to come into play at the site occupied by the actual group such as visual and auditory input of others, physical interaction between animals, release of additional stress pheromones etc., one cannot replace the amount of pheromone extracts collected at one point with group size. The importance of the study does not fade at all by describing the fact more faithfully, and in fact, would rather increase. I formed a very different expectation about the study at the beginning as the term "group size" already appears in the title and the abstract. This term should be used to place the motivation and finding into a big context only in part of the introduction and discussion.

Response 2.1: We are grateful to both reviewers for pointing out this central issue. We agree that we were missing a crucial step to be able to make those claims, namely that pheromone deposits act as a proxy for group size. We have addressed this issue by doing additional experiments, and re-writing parts of this manuscripts. We kindly refer reviewer #2 to the detailed **response 1.2** to a similar question by reviewer #1.

MINOR COMMENTS

Comment 2.2: Line 92: “We also hypothesized that group size indicators may be volatile, giving individuals the opportunity to assess the group size of several sites before joining.” / Line 105: “We also expect that this decision should be made prior to joining the group.”

The term join implies that once a fly decides to go into the group, it will remain as a member for an extended period. Has this been systematically investigated? Please provide the literature describing the behavior.

Response 2.2: The question of how long a fly stays in a group on a particular substrate is difficult to answer considering the lack of experimental studies on this aspect of *Drosophila*'s behavioral ecology. There is nevertheless a recent paper that observed flies in nature and found that the number of flies that arrives hourly on a fruit is higher than the number that leaves (Dukas, 2020). In our new experiments (Figure 1A), we measured the amount of pheromones deposited by flies occupying a glass vial for 90 minutes, taking 90 minutes as a reasonable amount of time for a fly to reside in a group based on the aforementioned field observation.

Comment 2.3: Line 273: “These results suggest that Or65a and Or67d may determine group size at different doses of cVA, and that they together determine the shape of the dose-response curve found for increasing doses of pheromone extracts.”

This can be relatively easily examined experimentally by expressing Kir using a combination of two Gal4 lines.

Response 2.3: We had performed this experiment during the pandemic lockdowns but discovered that our recombinant between Or67a and Or67d was contaminated. We thus unfortunately had to remove this from the manuscript.

Comment 2.4: Line 355: “The requirement for the simultaneous sensing of these two pheromones prevents mated females from joining groups consisting of virgin females.”

There is no evidence that sensing of two pheromones “prevents” attraction.

Response 2.4: We agree with the reviewer that the flies are not prevented, but instead not attracted. On page 12/line 381, we changed the text to “The combination of cVA and 7-T is thus a specific cue of the presence of flies that are good indicators of egg-laying site quality and means that mated females are not attracted to joining groups consisting of virgin females.”.

Comment 2.5: Line 388: “whereas Or65a seems necessary for the inhibition when groups become too large.”

This does not seem to match the results shown in Fig. 4C.

Response 2.5: Our previous statement was indeed a misinterpretation of the visualized results. On Page 13/Line 427, we have reformulated this sentence to ‘Our results suggest that Or67d is involved in the attraction to small groups, whereas Or65a seems necessary for mediating the response to groups that become larger’. We think that this reformulation does better justice to the visualized data. Larger groups are expected to become repellent, yet with a functioning Or65a receptor the flies are less discouraged from joining larger groups than when Or65a is not functioning. Without the involvement of Or65a, the response to the largest group size is more negative, hence we interpreted this as a mediating effect.

Comment 2.6: Line 430: “Considering that Or22a ORN is necessary for the response to 7-T (Fig. 5), it is conceivable that heptanal is perceived as a food cue rather than strictly a social cue in the context of group size determination.”

Given this, there is a possibility that food odor-sensitive ORNs that are not examined in this study are also involved in the behavior examined in the study. It might be worth discussing this issue.

Response 2.6: Many thanks for raising this point, which was also noted by reviewer #1. Please refer to responses 1.4 and 1.8 for a description of how we address this point in the revised version of this manuscript.

Comment 2.7: *Fig. 2B: It was difficult to interpret the labels. If the four labels at the bottom describe the genotype of the depositor, what do Oe- (light purple) and w[1118] (dark purple) above those labels mean?*

Response 2.7: We had overlooked the confusing labeling in figure 2B and thank the reviewer for bringing this to our attention. We have adjusted this by removing the legend boxes and including the genotypes in the axis text instead.

Reviewers' Comments:

Reviewer #1:

Remarks to the Author:

Re-Response 1.2 The authors took great pains to provide additional experiments showing that the concentration of pheromones increases with group size, including testing groups of males. However, they curiously, and at this point suspiciously, do not perform the simple experiment of directly testing groups in this study to verify that, as their title indicates: "Resident group size modulates oviposition site selection in *Drosophila melanogaster*".

Essentially, the study uses the following logic:

- a) Pheromones increase with group size
- b) Pheromones impact oviposition
- c) Therefore, group size affects oviposition

This is false chain reasoning or an example of invalid reasoning where "If P, then Q; If P, then R; Therefore: If Q, then R."

One cannot assume Group size necessarily impacts oviposition in the same way that:

Cats are mammals
Cats are predators
Therefore mammals are predators
is not correct.

Unfortunately, if the authors refuse to either adjust their logic or otherwise attempt the straightforward experiment of showing directly that groups do impact oviposition, this paper is fundamentally logically flawed. One simply cannot use pheromones as a "proxy for group size" without ensuring that there are NO other aspects of groups that might otherwise impact the results. This can only be assured by testing actual groups of animals themselves. This fatal flaw is a pity as the experiments are quite nice otherwise. I encourage the authors to temper their conclusions and rather make the more straightforward argument that increasing pheromone deposition impacts oviposition in the absence of fly presence.

Re-Response 1.7 vs. 1.10 According to the authors, Figure 1 shows that oviposition preference decreases with increasing fly equivalents on a substrate with yeast. Yet the authors argue that the oviposition preference should "increase up to a group size and then drop, forming a non-linear inverted U-shape curve." but this only happens with a poor substrate, which their previous work shows that females do not prefer to oviposit on. Thus, the whole argument seems very confusing,

Technical issue: Re-Response 1.11 and 1.12 As a chemist and chemical ecologist, I can confirm that hexane residue can build up on a substrate even after using nitrogen. Thus, the authors should control for this build up since they are adding more solvent with increasing fly equivalents.

Reviewer #2:

Remarks to the Author:

I appreciate the authors' additional effort to show that the amount of pheromone deposited in a glass tube in 90 min increases as a function of the number of flies enclosed (Fig. 1B), which is to be expected, this relationship does not allow the authors to conclude that the fly group size determines the selection of oviposition site. In all the rest of the experiment, flies were asked to choose between two cups filled with different concentration of pheromones. As there are no flies in this situation, it is a simple two alternative chemical choice task. This may mimic a situation where a single fly selects

between two areas where two groups of flies had spent some time in the past but are currently completely gone. As I have pointed out during the initial round of review, many more factors besides pheromone concentration are known to come into play at the site occupied by the actual group such as visual and auditory input of others, physical interaction between animals, release of additional stress pheromones etc. The results presented in the manuscript are solid and convincing, but they do not control for these factors, and thus do not support that "Resident groups size modulates oviposition site selection", the title and the main conclusion of the study. I cannot recommend its publication unless a major textual revision is made on this issue.

DETAILED ANSWERS TO REVIEWERS' COMMENTS

We thank both reviewers for this second round of feedback. It is clear from both their reviews that they disagree with our usage of the term “resident group”, and encourage us instead to focus on the effect of pheromonal dose. We have taken their comments fully onboard and changed the title of the manuscript from “Resident group size modulates oviposition site selection in *Drosophila melanogaster*” to “Aggregation pheromones have a non-linear effect on communal oviposition in *Drosophila melanogaster*”, as well as extensively edited the text. The edits are substantial and documented in a manuscript with changes tracked, followed by a clean version without changes tracked. These edits do not radically change our conclusions or experimental approach, but clarify our logic and removes controversial and -we agree- confusing statements about “resident groups”.

Please find below a detailed response to the comments. We have highlighted in Yellow the part of the review we specifically answered. We hope the reviewers will find that the advice they gave us is reflected in this manuscript with revised title and text.

REVIEWER #1 (REMARKS TO THE AUTHOR):

Re-Response 1.2 The authors took great pains to provide additional experiments showing that the concentration of pheromones increases with group size, including testing groups of males. However, they curiously, and at this point suspiciously, do not perform the simple experiment of directly testing groups in this study to verify that, as their title indicates: “Resident group size modulates oviposition site selection in *Drosophila melanogaster*”.

Essentially, the study uses the following logic:

- a) Pheromones increase with group size
- b) Pheromones impact oviposition
- c) Therefore, group size affects oviposition

This is false chain reasoning or an example of invalid reasoning where “If P, then Q; If P, then R; Therefore: If Q, then R.”

One cannot assume Group size necessarily impacts oviposition in the same way that:

Cats are mammals
Cats are predators
Therefore mammals are predators
is not correct.

Unfortunately, if the authors refuse to either adjust their logic or otherwise attempt the straightforward experiment of showing directly that groups do impact oviposition, this paper is fundamentally logically flawed. One simply cannot use pheromones as a “proxy for group size” without ensuring that there are NO other aspects of groups that might otherwise impact the results. This can only be assured by testing actual groups of animals themselves. This fatal flaw is a pity as the experiments are quite nice otherwise. I encourage the authors to temper their conclusions and rather make the more straightforward argument that increasing pheromone deposition impacts oviposition in the absence of fly presence.

Response 1.1: We apologize for the confusion we created by using the term “resident group” and thank both reviewers for highlighting that this term is not in keeping with the logic of our experiment. We have now heavily edited the manuscript to clarify our logic. The reviewer is of course correct in stating that the actual group size of the adults could be sensed with different sensory modalities. Our focus, however, has been on dissecting the volatile chemical cues that females use to infer how many adults have been utilizing a communal oviposition site. This allows the females to make a quantitative estimation on the resident egg/larval/adult group size to which they would be contributing their eggs. As we now explain in the introduction, this is relevant, as natural oviposition sites are used over the course of several days. Those females that have been there before cannot be seen anymore, but are relevant; we revealed that they can be smelled and that this influence oviposition site choice in a non-linear manner. So, to clarify, our aim was to test

whether the females can use pheromonal components that provide quantitative information on the presence of others that have contributed to the communal oviposition site, and how females use that quantitative information for their oviposition decisions. Of course, also non-volatiles stimuli (e.g., gustatory or visual cues on the presence of other females, eggs or larvae) could be sensed and perceived, and may contribute to the oviposition behavior of females. Yet, these additional factors are not the focus of our investigations, and we designed our bio-assay specifically so that only chemical volatile cues could be perceived.

We have changed the title to “Aggregation pheromones have a non-linear effect on communal oviposition in *Drosophila melanogaster*” and removed any presumption that the oviposition choice we report reflects behaviour in, or towards, an *in situ* group of adult flies. Practically, this resulted in completely removing the concept of “resident group” throughout the manuscript and instead focusing on the response to pheromones, as advised by the reviewers. We emphasize - following reviewer #1’s recommendation - that flies respond to the presence of increasing pheromones and explain that those are left by prior resident adult flies: it is not about who is there, but who has been there. As we now explain in the introduction, this is relevant, as natural oviposition sites are used over the course of several days. Those that have been there before cannot be seen anymore, but are relevant; we revealed that they can be smelled and that this influence oviposition site choice in a non-linear manner.

Re-Response 1.7 vs. 1.10 According to the authors, Figure 1 shows that oviposition preference decreases with increasing fly equivalents on a substrate with yeast. Yet the authors argue that the oviposition preference should “increase up to a group size and then drop, forming a non-linear inverted U-shape curve.” but this only happens with a poor substrate, which their previous work shows that females do not prefer to oviposit on. Thus, the whole argument seems very confusing,

Response 1.2: We have now further explained the theoretical basis for this non-linear response in the absence of food.

We developed our explanation -based on published articles- of the theoretical basis for the decrease in importance of pheromone in the presence of food on line 141: “Communal oviposition enhances oviposition site quality, both by the inoculation of yeasts, acting as a larval food source, by the adults, and because groups of larvae are better at reducing the hyphal growth of molds that compete for food with the larvae ^{22, 23, 24}. However, strong resource competition ^{25, 26} and increased attraction of natural enemies may occur when groups of larvae become too large ^{16, 27}. Modeling of population persistence and female decision making, based on behavioral and fecundity data of *Drosophila* females, predicted that substrate quality and adult group density are main drivers for aggregation and communal oviposition ^{28, 29}. Based upon these findings, it is expected that positive density-dependent effects occur when groups of larvae are neither too small nor too large ^{10, 11}. Hence, females would benefit from assessing the number of females who have (previously) contributed to a communal oviposition site before deciding on adding their own eggs to that site. A mechanism for this would be the ability to sense the dose of chemical cues left by contributors at the oviposition site to estimate whether a positive density-dependent effect on larval fitness may occur at a communal oviposition site.”

And explain our predictions based on line 240: “In this study, we investigate whether and how females use pheromones as quantitative cue to determine the suitability of communal oviposition sites. We hypothesize that the females’ oviposition decisions are modulated by the dose of volatile pheromones, as these provide an indication for the number of individuals that have contributed to that site. Since larval survival at communal oviposition sites follows a hump-shaped relationship with larval density, in which sites with too few or too many larvae will impose developmental constraints ^{11, 22, 24, 52}, we hypothesize that the attraction to pheromonal deposits will increase up to

an optimum and then decrease. Given that flies assess both social and nutritional cues when determining the quality of an oviposition sites for larval survival²⁰, we expect that the decision to oviposit at a communal oviposition site is based on cues from the number of visitors to the communal site and cues about the nutritional quality. We also expect that long-distance cues are guiding the decision to visit the oviposition site. Therefore, we focused on olfactory cues and not on gustatory cues, which would necessitates micro-assessment of the substrate once arrived on the substrate^{47, 48, 53}. Finally, considering that the occurrence of positive density dependent effects are expected to depend on resource conditions in relation to group size^{10, 11, 20, 28}, we hypothesize that females rely less on pheromonal cues when evaluating oviposition substrates of high nutritional quality than on substrates of low nutritional quality, where larvae depend more on cooperative behavior to survive on poor nutrition substrates”.

Technical issue: Re-Response 1.11 and 1.12 As a chemist and chemical ecologist, I can confirm that hexane residue can build up on a substrate even after using nitrogen. Thus, the authors should control for this build up since they are adding more solvent with increasing fly equivalents.

Thank you for this comment. Besides the swift evaporation of the hexane, we have controlled for this by pipetting the equivalent of hexane - used to load the phenomes – as a control treatment. As a result, the two cups should contain an equivalent of hexane build up and serve as a control for the increasing quantities of hexane. In our design, the pheromones are loaded within a small mesh covered cup that doesn't allow the flies to get into contact with any hexane residue left behind on the substrate. We assume that this also minimizes the effects that any potential hexane residue could have had on the flies.

Response 1.2:

REVIEWER #2 (REMARKS TO THE AUTHOR):

I appreciate the authors' additional effort to show that the amount of pheromone deposited in a glass tube in 90 min increases as a function of the number of flies enclosed (Fig. 1B), which is to be expected, this relationship does not allow the authors to conclude that the fly group size determines the selection of oviposition site. In all the rest of the experiment, flies were asked to chose between two cups filled with different concentration of pheromones. As there are no flies in this situation, it is a simple two alternative chemical choice task. This may mimic a situation where a single fly selects between two areas where two groups of flies had spent some time in the past but are currently completely gone. As I have pointed out during the initial round of review, many more factors besides pheromone concentration are known to come into play at the site occupied by the actual group such as visual and auditory input of others, physical interaction between animals, release of additional stress pheromones etc. The results presented in the manuscript are solid and convincing, but they do not control for these factors, and thus do not support that “Resident groups size modulates oviposition site selection”, the title and the main conclusion of the study. I cannot recommend its publication unless a major textual revision is made on this issue.

Response 2.1: This review independently states the same concern as that of reviewer #1. We thank this reviewer for pushing us to clarify our findings and to remove confusing arguments about resident group size. As detailed in response 1.1, we have changed the title and extensively edited the text to clarify our logic, including a clarification that pheromones are left by visitors of a site, and not current residents. This clarifies why visual, auditory and physical interactions, which are associated with individual that are physically present at a site, are not part of our hypotheses and not investigated in this manuscript.

Reviewers' Comments:

Reviewer #1:

Remarks to the Author:

The authors have indeed edited the manuscript significantly. As stated by both reviewers in previous reviews, the study presented has shown convincingly that female oviposition rates are altered by the presence of different concentrations of fly-released chemicals in the substrate (cVA and heptanal). The manuscript also provides an intriguing hypothesis that these cues could be significant (density-dependent) survival fitness indicators for the importance of community in the case of poor substrates.

Indeed, this is an intriguing hypothesis and one worth exploring, as it has been in other systems. Unfortunately, while the authors adjusted some language, the manuscript still insists, particularly in the discussion, that the dose-dependent response to these chemicals is an indicator of group size and that group size directly impacts oviposition. In fact, in many cases, the term "group size" is simply replaced with synonyms: "number of site visitors", "communal oviposition site", "aggregation", "social density", etc. Yet, as stated by both reviewers on multiple occasions, the direct impact of groups on oviposition has still not been shown in this manuscript and thus cannot be discussed.

The study presented has not shown that the concentration of cVA and heptanal indicates group size for the fly itself. The chemicals might be triggering some relationship to mating that impacts egg laying (I realize this is not as parsimonious as group size, but then again, it cannot be refuted with the current data). Secondly, the authors have not shown that groups themselves impact oviposition, which has been suggested by both reviewers previously.

In summary, the manuscript does not seem to improve through these rounds, and I no longer see a way forward. If the authors insist on discussing groups, the study requires experiments to test this hypothesis directly.

Reply to reviewer's comments

Reviewer #1 (Remarks to the Author):

The authors have indeed edited the manuscript significantly. As stated by both reviewers in previous reviews, the study presented has shown convincingly that female oviposition rates are altered by the presence of different concentrations of fly-released chemicals in the substrate (cVA and heptanal). The manuscript also provides an intriguing hypothesis that these cues could be significant (density-dependent) survival fitness indicators for the importance of community in the case of poor substrates.

Indeed, this is an intriguing hypothesis and one worth exploring, as it has been in other systems. Unfortunately, while the authors adjusted some language, the manuscript still insists, particularly in the discussion, that the dose-dependent response to these chemicals is an indicator of group size and that group size directly impacts oviposition. In fact, in many cases, the term "group size" is simply replaced with synonyms: "number of site visitors", "communal oviposition site", "aggregation", "social density", etc. Yet, as stated by both reviewers on multiple occasions, the direct impact of groups on oviposition has still not been shown in this manuscript and thus cannot be discussed.

The study presented has not shown that the concentration of cVA and heptanal indicates group size for the fly itself. The chemicals might be triggering some relationship to mating that impacts egg laying (I realize this is not as parsimonious as group size, but then again, it cannot be refuted with the current data). Secondly, the authors have not shown that groups themselves impact oviposition, which has been suggested by both reviewers previously.

In summary, the manuscript does not seem to improve through these rounds, and I no longer see a way forward. If the authors insist on discussing groups, the study requires experiments to test this hypothesis directly.

Response: With the help of the editor, the revised version of our manuscript now clarifies the limitations of our study and highlights conclusions which are in keeping with this reviewer's summary of our main findings. The editor carefully indicated sentences that might make too strong a claim, and we have accepted all these comments.

In addition, we make it clear -a point important to both reviewers- that we did not directly test groups of adult flies. We have added alternative explanation and limitations to our study in the introductory paragraph of the discussion.

lines (355-361) "Because we aimed to specifically study the olfactory social component of attraction to egg-laying sites, a limitation of our study is that we did not test attraction to different group sizes of adult flies, just the pheromones they can deposit. It is likely that additional social factors modulate oviposition site selection once flies have arrived at an oviposition site. For example, flies physically present at the oviposition site may modify oviposition behavior through visual input ⁷¹, physical interaction between individuals ⁷², or the release of stress pheromones ^{73,74}."

We added the following sentences, lines 348-352 "Because we tested the attraction of females towards oviposition substates marked by pheromone extracts from different numbers of individuals instead of testing attraction to the individuals themselves, it is important to note that our results uncover a mechanism potentially allowing females to determine the number of prior individuals sharing a communal oviposition site through olfaction alone."

AS for the comment "The chemicals might be triggering some relationship to mating that impacts egg laying (I realize this is not as parsimonious as group size, but then again, it cannot be refuted with the current data).

If we understood the proposition correctly, we agree that this not a very parsimonious explanation but nonetheless interesting and worth mentioning. We have added the following: on line 384-397 (clean version). "The combination of cVA and 7-T is also involved in mate-guarding, by making females resemble the pheromone profile of males, which inhibits courtship

attempts by males through physical contact with taste receptor Gr32a⁵⁴. Because of their use in sexual communication, females might preferentially lay eggs on sites containing intermediate amount of these pheromones, not because they are evaluating it as a preferred place to lay their eggs, but because it could be a good place to mate. An argument against this interpretation is that mated females have reduced sexual receptivity compared to virgin females and that virgin females are not attracted to egg-laying sites marked with pheromones¹⁸. It is thus unlikely that the primary driver of mated female laying eggs on substrate with intermediate pheromone concentration is to obtain mates. Although it relies on the same pheromones, our findings suggest a different sensory pathway is involved in oviposition site attraction than in mating. In the case of oviposition site attraction, it is the oxidation product of 7-T - heptanal detected through olfaction that attracts females to oviposition sites in our study. This illustrates how different sensory modalities detecting the same cues shape context-dependent behavioral responses^{46,48}.”